# Visual Programming for Text-to-Image Generation and Evaluation

**Jaemin Cho**      **Abhay Zala**      **Mohit Bansal**
UNC Chapel Hill
{jmincho, aszala, mbansal}@cs.unc.edu
https://vp-t2i.github.io

## Abstract

As large language models have demonstrated impressive performance in many domains, recent works have adopted language models (LMs) as controllers of visual modules for vision-and-language tasks. While existing work focuses on equipping LMs with visual understanding, we propose two novel interpretable/explainable visual programming frameworks for text-to-image (T2I) generation and evaluation. First, we introduce VPGEN, an interpretable step-by-step T2I generation framework that decomposes T2I generation into three steps: object/count generation, layout generation, and image generation. We employ an LM to handle the first two steps (object/count generation and layout generation), by finetuning it on text-layout pairs. Our step-by-step T2I generation framework provides stronger spatial control than end-to-end models, the dominant approach for this task. Furthermore, we leverage the world knowledge of pretrained LMs, overcoming the limitation of previous layout-guided T2I works that can only handle predefined object classes. We demonstrate that our VPGEN has improved control in counts/spatial relations/scales of objects than state-of-the-art T2I generation models. Second, we introduce VPEVAL, an interpretable and explainable evaluation framework for T2I generation based on visual programming. Unlike previous T2I evaluations with a single scoring model that is accurate in some skills but unreliable in others, VPEVAL produces evaluation programs that invoke a set of visual modules that are experts in different skills, and also provides visual+textual explanations of the evaluation results. Our analysis shows that VPEVAL provides a more human-correlated evaluation for skill-specific and open-ended prompts than widely used single model-based evaluation. We hope that our work encourages future progress on interpretable/explainable generation and evaluation for T2I models.

## 1  Introduction

Large language models (LLMs) have shown remarkable performance on many natural language processing tasks, such as question answering, summarization, and story generation [1; 2; 3; 4; 5; 6; 7; 8]. Recent works have shown that LLMs can also tackle certain vision-and-language tasks such as visual question answering and visual grounding, by generating visual programs that can control external visual modules and combine their outputs to get the final response [9; 10; 11; 12; 13; 14; 15]. However, no prior works have combined LLMs and different visual modules for the challenging text-to-image (T2I) generation task. Our work proposes two novel interpretable/explainable visual programming (VP) frameworks combining LLMs and visual modules for T2I generation and evaluation.

First, we introduce **VPGEN** (Sec. 3), a new step-by-step T2I generation framework that decomposes the T2I task into three steps (object/count generation, layout generation, and image generation), where each step is implemented as a module and executed in an interpretable generation program, as shown

37th Conference on Neural Information Processing Systems (NeurIPS 2023).

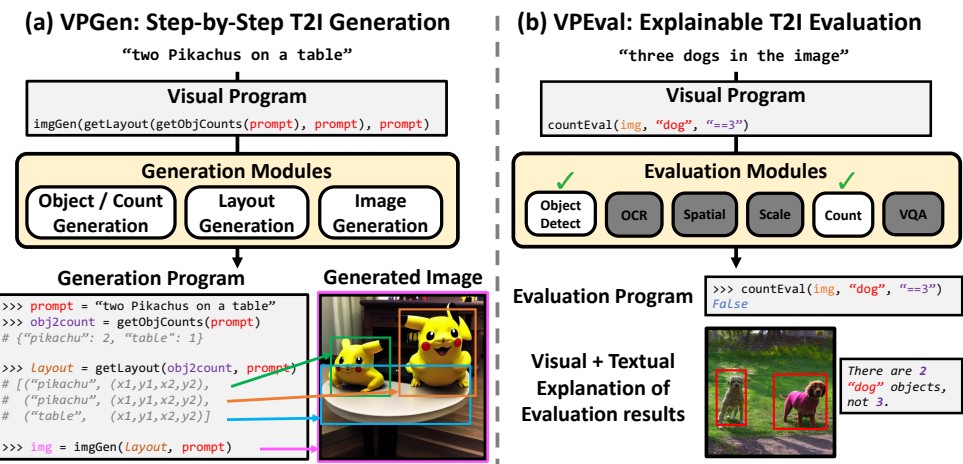

Figure 1: Illustration of the proposed visual programming frameworks for text-to-image (T2I) generation and evaluation. In (a) VPGEN, we first generate a list of objects, then object positions, and finally an image, by executing three modules step-by-step. In (b) VPEVAL, we use evaluation programs with a mixture of evaluation modules that handle different skills and provide visual+textual explanation of evaluation results.

in Fig. 1 (a). We employ Vicuna [16], a powerful publicly available LLM, to handle the first two steps (object/count generation and layout generation), by finetuning it on text-layout pairs from multiple datasets [17; 18; 19], resulting in improved layout control for T2I generation. For the last image generation step, we use off-the-shelf layout-to-image generation models, *e.g.*, GLIGEN [20]. Our generation framework provides more interpretable spatial control than the widely used end-to-end T2I models [21; 22; 23; 24]. Furthermore, VPGEN leverages the world knowledge of pretrained LMs to understand unseen objects, overcoming the limitation of previous layout-guided T2I works that can only handle predefined object classes [25; 26; 27; 28]. In our analysis, we find that our VPGEN (Vicuna+GLIGEN) could generate images more accurately following the text description (especially about object counts, spatial relations, and object sizes) than strong T2I generation baselines, such as Stable Diffusion [21] (see Sec. 5.2).

Next, we introduce **VPEVAL** (Sec. 4), a new interpretable/explainable T2I evaluation framework based on evaluation programs that invoke diverse visual modules to evaluate different T2I skills [19] (see Sec. 4.1), and also provide visual+textual explanations of the evaluation results. Previous T2I evaluation works have focused on measuring visual quality [29; 30] and image-text alignment with a single visual module based on text retrieval [31], CLIP cosine similarity [32; 33], captioning [25], object detection [34; 19; 35], and visual question answering (VQA) [36; 37]. However, we cannot interpret the reasoning behind the scoring; *i.e.*, why CLIP assigns a higher score to an image-text pair than another. In addition, these modules are good at measuring some skills, but not reliable in other skills (*e.g.*, VQA/CLIP models are not good at counting objects or accurately parsing text rendered in images). In VPEVAL, we break the evaluation process into a mixture of visual evaluation modules, resulting in an interpretable evaluation program. The evaluation modules are experts in different skills and provide visual+textual (*i.e.*, multimodal) result/error explanations, as shown in Fig. 1 (b). We evaluate both multimodal LM and diffusion-based T2I models with two types of prompts: (1) skill-based prompts (Sec. 4.3) and (2) open-ended prompts (Sec. 4.4). Skill-based prompts evaluate a single skill per image, whereas open-ended prompts evaluate multiple skills per image. We adapt a large language model (GPT-3.5-Turbo [38]) to dynamically generate an evaluation program for each open-ended prompt, without requiring finetuning on expensive evaluation program annotations.

In our skill-based prompt experiments, while all T2I models faced challenges in the Count, Spatial, Scale, and Text Rendering skills, our VPGEN shows higher scores in the Count, Spatial, and Scale skills, demonstrating its strong layout control. We also show a fine-grained sub-split analysis on where our VPGEN gives an improvement over the baseline T2I models. In our open-ended prompt experiments, our VPGEN achieves competitive results to the T2I baselines. Many of these open-ended prompts tend to focus more on objects and attributes, which, as shown by the skill-based evaluation, T2I models perform quite well in; however, when precise layouts and spatial relationships are needed, our VPGEN framework performs better (Sec. 5.3). In our error analysis, we also find

that GLIGEN sometimes fails to properly generate images even when Vicuna generates the correct layouts, indicating that as better layout-to-image models become available, our VPGEN framework will achieve better results (Sec. 5.3). We also provide an analysis of the generated evaluation programs from VPEVAL, which proved to be highly accurate and comprehensive in covering elements from the prompts (Sec. 5.4). For both types of prompts, our VPEVAL method shows a higher correlation to human evaluation than existing single model-based evaluation methods (Sec. 5.4).

Our contributions can be summarized as follows: (1) **VPGEN** (Sec. 3), a new step-by-step T2I generation framework that decomposes the T2I task into three steps (object/count generation, layout generation, and image generation), where each step is implemented as a module and executed in an interpretable generation program; (2) **VPEVAL** (Sec. 4), a new interpretable/explainable T2I evaluation framework based on evaluation programs that execute diverse visual modules to evaluate different T2I skills and provide visual+textual explanations of the evaluation results; (3) comprehensive analysis of different T2I models, which demonstrates the strong layout control of VPGEN and high human correlation of VPEVAL (Sec. 5). We will release T2I model-generated images, VPEVAL programs, and a public LM (finetuned for evaluation program generation using ChatGPT outputs). We hope our research fosters future work on interpretable/explainable generation and evaluation for T2I tasks.

## 2   Related Works

**Text-to-image generation models.** In the T2I generation task, models generate images from text. Early deep learning models used the Generative Adversarial Networks (GAN) [39] framework for this task [40; 41; 42; 31]. More recently, multimodal language models [43; 22] and diffusion models [44; 45; 21; 46] have gained popularity. Recent advances in multimodal language models such as Parti [47] and MUSE [48], and diffusion models like Stable Diffusion [21], UnCLIP [23], and Imagen [24], have demonstrated a high level of photorealism in zero-shot image generation.

**Bridging text-to-image generation with layouts.** One line of research decomposes the T2I generation task into two stages: text-to-layout generation and layout-to-image generation [25; 26; 27; 28]. However, the previous approaches focus on a set of predefined object classes by training a new layout predictor module from scratch and therefore cannot place new objects unseen during training. In contrast, our VPGEN uses an LM to handle layout generation by generating objects/counts/positions in text, allowing flexible adaptation of pretrained LMs that can understand diverse region descriptions.

**Language models with visual modules.** Although large language models (LLMs) have shown a broad range of commonsense knowledge, most of them are trained only on text corpus and cannot understand image inputs to tackle vision-and-language (VL) tasks. Thus, recent works explore tackling VL tasks by solving sub-tasks with external visual modules and combining their outputs to obtain the final response [9; 10; 11; 12; 13; 14; 15]. The visual sub-tasks include describing images as text, finding image regions relevant to the text, editing images with text guidance, and obtaining answers from a VQA model. However, existing work focuses on converting visual inputs into text format so that LLMs can understand them. Our work is the first work using visual programming for interpretable and explainable T2I generation and evaluation.

**Evaluation of text-to-image generation models.** The text-to-image community has commonly used two types of automated evaluation metrics: visual quality and image-text alignment. For visual quality, Inception Score (IS) [29] and Fréchet Inception Distance (FID) [30] have been widely used. For image-text alignment, previous work used a single model to calculate an alignment score for image-text pair, based on text retrieval [31], cosine similarity [33], captioning [25], object detection [34; 19; 35], and visual question answering (VQA) [36; 37]. In this work, we propose the first T2I evaluation framework VPEVAL, based on interpretable and explainable evaluation programs which execute a diverse set of visual modules (*e.g.*, object detection, OCR, depth estimation, object counting). Our VPEVAL evaluation programs provide visual+textual explanations of the evaluation result and demonstrate a high correlation with human judgments.

## 3   VPGEN: Visual Programming for Step-by-Step Text-to-Image Generation

We propose VPGEN, a novel visual programming framework for interpretable step-by-step text-to-image (T2I) generation. As illustrated in Fig. 2, we decompose the text-to-image generation task into

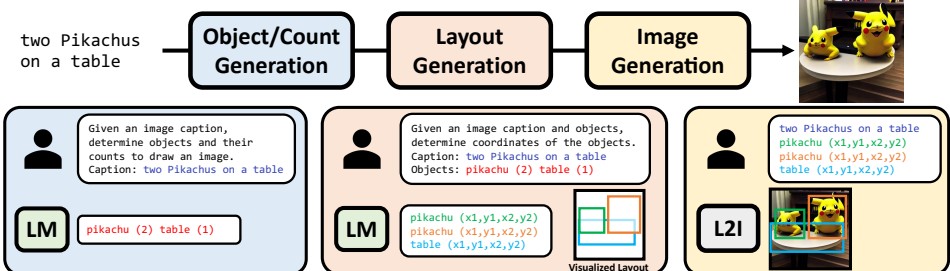

Figure 2: Interpretable step-by-step text-to-image generation with VPGEN. VPGEN decomposes T2I generation into three steps: (1) object/count generation, (2) layout generation, and (3) image generation, and executes the three modules step-by-step.

three steps: (1) object/count generation, (2) layout generation, and (3) image generation. In contrast to previous T2I generation works that use an intermediate layout prediction module [25; 26; 27; 28], VPGEN represents all layouts (object description, object counts, and bounding boxes) in text, and employs an LM to handle the first two steps: (1) object/count generation and (2) layout generation. This makes it easy to adapt the knowledge of pretrained LMs and enables generating layouts of objects that are unseen during text-to-layout training (*e.g.*, 'pikachu'). For layout representation, we choose the bounding box format because of its efficiency; bounding boxes generally require fewer tokens than other formats.[1]

**Two-step layout generation with LM.** Fig. 2 illustrates how our LM generates layouts in two steps: (1) object/count generation and (2) layout generation. For the first step, we represent the scene by enumerating objects and their counts, such as ''obj1 (# of obj1) obj2 (# of obj2)''. For the second step, following previous LM-based object detection works [49; 50], we normalize 'xyxy' format bounding box coordinates into $[0, 1]$ and quantize them into 100 bins; a single object is represented as ''obj (xmin,ymin,xmax,ymax)'', where each coordinate is within $\{0, \cdots, 99\}$.

**Training layout-aware LM.** To obtain the layout-aware LM, we use Vicuna 13B [16], a public state-of-the-art language model finetuned from LLaMA [51]. We use parameter-efficient finetuning with LoRA [52] to preserve the original knowledge of the LM and save memory during training and inference. We collect text-layout pair annotations from training sets of three public datasets: Flickr30K entities [17], MS COCO instances 2014 [18], and PaintSkills [19], totaling 1.2M examples. See appendix for more training details.

**Layout-to-Image Generation.** We use a recent layout-to-image generation model GLIGEN [20] for the final step - image generation. The layout-to-image model takes a list of regions (bounding boxes and text descriptions) as well as the original text prompt to generate an image.

## 4 VPEVAL: Visual Programming for Explainable Evaluation of Text-to-Image Generation

VPEVAL is a novel interpretable/explainable evaluation framework for T2I generation models, based on visual programming. Unlike existing T2I evaluation methods that compute image-text alignment scores with an end-to-end model, our evaluation provides an interpretable program and visual+textual explanations for the evaluation results, as shown in Figs. 3 and 5. We propose two types of evaluation prompts: (1) skill-based evaluation and (2) open-ended evaluation. In skill-based evaluation, we define five image generation skills and use a set of skill-specific prompts and evaluation programs, as illustrated in Fig. 3. In open-ended evaluation, we use a diverse set of prompts that require multiple image generation skills. We adopt a language model to dynamically generate an evaluation program for each text prompt, as shown in Fig. 5. In the following, we describe evaluation skills (Sec. 4.1), visual evaluation modules (Sec. 4.2), skill-based evaluation with visual programs (Sec. 4.3), and open-ended evaluation with visual program generator LM (Sec. 4.4).

---

[1]For example, a xyxy-format bounding box can be represented with 4 tokens, while a 64x64 segmentation map requires 4096 tokens.

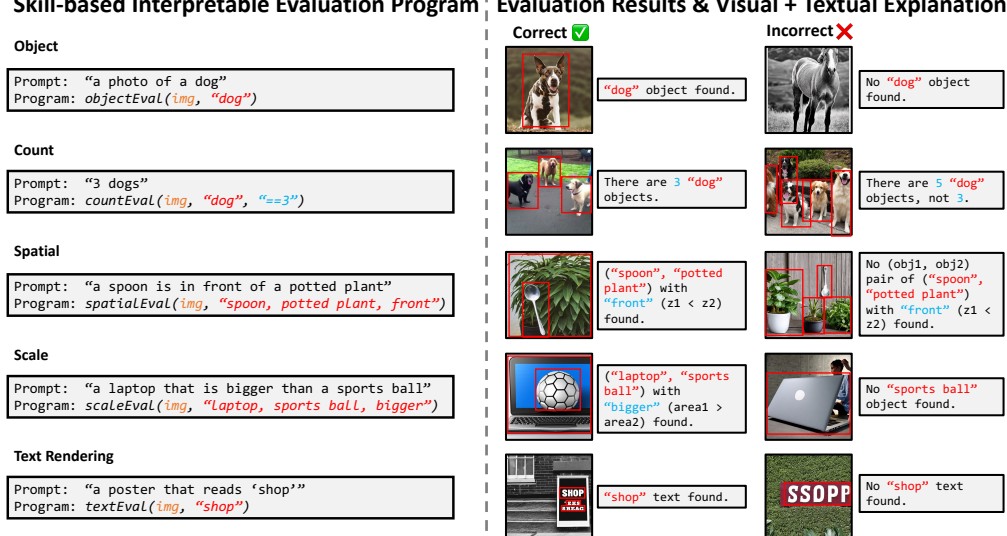

Figure 3: Illustration of skill-based evaluations in VPEVAL (Sec. 4.3). **Left**: Given text prompts that require different image generation skills, our interpretable evaluation programs evaluate images by executing relevant visual modules. **Right**: Our evaluation programs output binary scores and provide visual (bounding boxes of detected objects/text) + textual explanations of the evaluation results.

## 4.1 Evaluation Skills

Inspired by skill-based T2I analysis of PaintSkills [19], our VPEVAL measures five image generation skills: Object, Count, Spatial, Scale, and Text Rendering. Powered by evaluation programs with expert visual modules, our VPEVAL supports zero-shot evaluation of images (no finetuning of T2I models is required), detecting regions with free-form text prompts, new 3D spatial relations (front, behind), and new scale comparison and text rendering skills, which were not supported in PaintSkills [19]. In Fig. 3, we illustrate the evaluation process for each skill.

**Object.** Given a prompt with an object (*e.g.*, "a photo of a dog"), a T2I model should generate an image with that object present.

**Count.** Given a prompt with a certain number of an object (*e.g.*, "3 dogs"), a T2I model should generate an image containing the specified number of objects.

**Spatial.** Given a prompt with two objects and a spatial relationship between them (*e.g.*, "a spoon is in front of a potted plant"), a T2I model should generate an image that contains both objects with the correct spatial relations.

**Scale.** Given a prompt with two objects and a relative scale between them (*e.g.*, "a laptop that is bigger than a sports ball"), a T2I model should generate an image that contains both objects, and each object should be of the correct relative scale.

**Text Rendering.** Given a prompt with a certain text to display (*e.g.*, "a poster that reads 'shop'"), a T2I model should generate an image that properly renders the text.

## 4.2 Visual Evaluation Modules

To measure the skills described above in Sec. 4.1, we use eight expert visual evaluation modules specialized for different tasks. The modules provide visual+textual explanations with their score. Visual explanations are generated by rendering bounding boxes on the images and textual explanations are generated through text templates, as shown in Fig. 3 and 5. We provide the Python pseudocode implementation for each module in Fig. 4.

**Module definitions. objDet** detects objects in an image based on a referring expression text and returns them (+ their 2D bounding boxes and depth), using Grounding DINO [53] and DPT [54]. **ocr** detects all text in an image and returns them (+ their 2D bounding boxes), using EasyOCR [55].

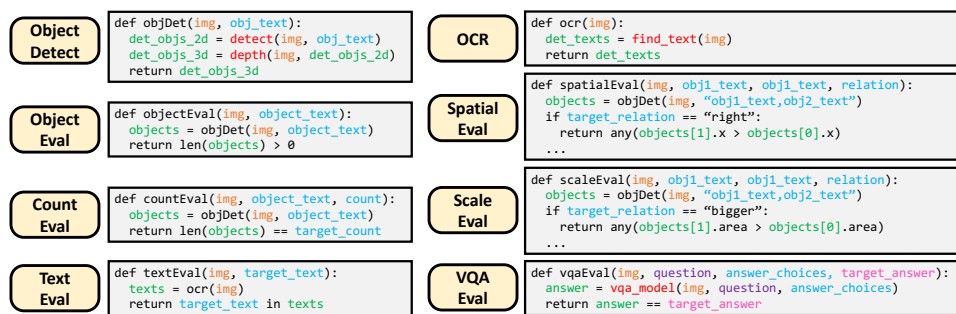

Figure 4: Python pseudocode implementation of visual modules used in VPEVAL.

**vqa** answers a multiple-choice question using BLIP-2 (Flan-T5 XL) [56]. It can handle phrases that cannot be covered only with objDet or ocr (*e.g.*, pose estimation, action recognition, object attributes). **objectEval** evaluates if an object is in an image using objDet. **countEval** evaluates if an object occurs in an image a certain number of times using objDet and an equation (*e.g.*, "==3", "<5"). **spatialEval** evaluates if two objects have a certain spatial relationship with each other. For six relations (above, below, left, right, front, and behind), it compares bounding box/depth values using objDet. For other relations, it uses vqa. **scaleEval** evaluates if two objects have a certain scale relationship with each other using objDet. For three scale relations (smaller, bigger, and same) it compares bounding box areas by using objDet. For other relations, it uses vqa. **textEval** evaluates if a given text is present in an image using ocr.

## 4.3 Skill-based Evaluation with Visual Programs

For skill-based evaluation, we create text prompts with various skill-specific templates that are used for image generation and evaluation with our programs. See appendix for the details of prompt creation. In Fig. 3, we illustrate our skill-based evaluation in VPEVAL. Given text prompts that require different image generation skills, our evaluation programs measure image-text alignment scores by calling the relevant visual modules. Unlike existing T2I evaluation methods, our evaluation programs provide visual+textual explanations of the evaluation results.

## 4.4 Open-ended Evaluation with Visual Program Generator LM

Although our evaluation with skill-specific prompts covers five important and diverse image generation skills, user-written prompts can sometimes be even more complex and need multiple evaluation criteria (*e.g.*, a mix of our skills in Sec. 4.1 and other skills like attribute detection). For example, evaluating images generated with the prompt 'A woman dressed for a wedding is showing a watermelon slice to a woman on a scooter.' involves multiple aspects, such as two women (count skill), 'a woman on a scooter' (spatial skill), 'dressed for a wedding' (attribute detection skill), *etc*. To handle such open-ended prompts, we extend the VPEVAL setup with evaluation programs that can use many visual modules together (whereas single-skill prompts can be evaluated with a program of 1-2 modules). We generate open-ended prompt evaluation programs with an LLM, then the evaluation programs output the average score and the visual+textual explanations from their visual modules. The program generation involves choosing which prompt elements to evaluate and which modules to evaluate those elements (see Fig. 5).

**Open-ended prompts.** For open-ended evaluation, we use 160 prompts of TIFA v1.0 human judgment dataset [36] (we refer to these prompts as 'TIFA160'). The dataset consists of (1) text prompts from COCO [18], PaintSkills [19], DrawBench [24], and Partiprompts [47], (2) images generated by five baseline models (minDALL-E [57], VQ-Diffusion [58], Stable Diffusion v1.1/v1.5/v2.1 [21]), and (3) human judgment scores on the images (on 1-5 Likert scale).

**Generating evaluation programs via in-context learning.** As annotation of evaluation programs with open-ended prompts can be expensive, we use ChatGPT (GPT-3.5-Turbo) [38] to generate evaluation programs via in-context learning. To guide ChatGPT, we adapt the 12 in-context prompts from TIFA [36]. We show ChatGPT the list of visual modules and example text prompts and programs, then ask the model to generate a program given a new prompt, as illustrated in Fig. 5. For

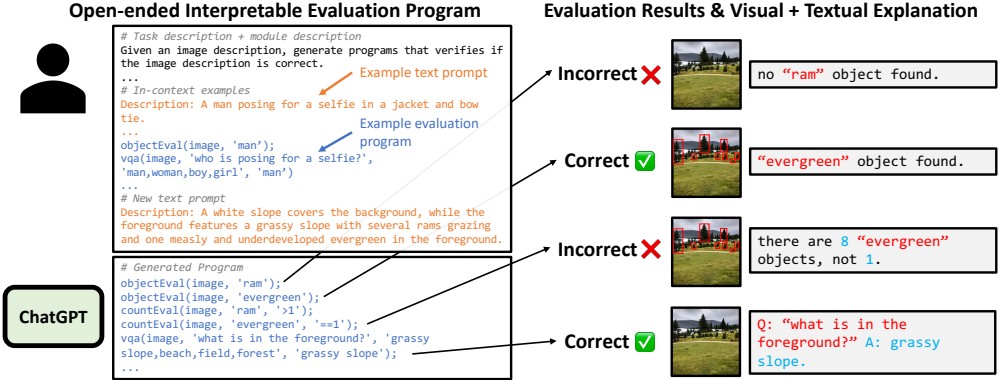

Figure 5: Evaluation of open-ended prompts in VPEVAL (Sec. 4.4). **Left**: We generate evaluation programs with ChatGPT via in-context learning. **Right**: Our evaluation programs consist of multiple modules evaluating different elements from a text prompt. We output the final evaluation score by averaging the outputs of each evaluation module.

Table 1: VPEVAL scores of T2I generation models on skill-based prompts (see Sec. 5.2 for analysis). *F30: Flickr30K Entities, C: COCO, P: PaintSkills*.

| Model | VPEVAL Skill Score (%) ↑ | | | | | |
|---|---|---|---|---|---|---|
| | Object | Count | Spatial | Scale | Text Rendering | Average |
| Stable Diffusion v1.4 | **97.3** | 47.4 | 22.9 | 11.9 | **8.9** | 37.7 |
| Stable Diffusion v2.1 | 96.5 | 53.9 | 31.3 | 14.3 | 6.9 | 40.6 |
| Karlo | 95.0 | 59.5 | 24.0 | 16.4 | **8.9** | 40.8 |
| minDALL-E | 79.8 | 29.3 | 7.0 | 6.2 | 0.0 | 24.4 |
| DALL-E Mega | 94.0 | 45.6 | 17.0 | 8.5 | 0.0 | 33.0 |
| VPGEN (F30) | 96.8 | 55.0 | 39.0 | 23.3 | 5.2 | 43.9 |
| VPGEN (F30+C+P) | 96.8 | **72.2** | **56.1** | **26.3** | 3.7 | **51.0** |

reproducible and accessible evaluation, we release the evaluation programs so that VPEVAL users do not have to generate the programs themselves. We will also release a public LM (finetuned for evaluation program generation using ChatGPT outputs) that can run on local machines.

# 5 Experiments and Results

In this section, we introduce the baseline T2I models we evaluate (Sec. 5.1), compare different T2I models with skill-based (Sec. 5.2) and open-ended prompts (Sec. 5.3), and show the human-correlation of VPEVAL (Sec. 5.4).

## 5.1 Evaluated Models

We evaluate our VPGEN (Vicuna+GLIGEN) and five popular and publicly available T2I models, covering both diffusion models (Stable Diffusion v1.4/v2.1 [21] and Karlo [59]), and multimodal autoregressive language models (minDALL-E [57] and DALL-E Mega [60]). Stable Diffusion v1.4 is the most comparable baseline to VPGEN, because the GLIGEN [20] in VPGEN uses frozen Stable Diffusion v1.4 with a few newly inserted adapter parameters for spatial control.

## 5.2 Evaluation on Skill-based Prompts

**Diffusion models outperform multimodal LMs.** In Table 1, we show the VPEVAL skill accuracies for each model. The diffusion models (Stable Diffusion, Karlo, and our VPGEN) show higher overall accuracy than the multimodal LMs (minDALL-E and DALL-E Mega).

**Count/Spatial/Scale/Text Rendering skills are challenging.** Overall, the five baseline T2I models achieve high scores (above 93% except for minDALL-E) in Object skill; *i.e.*, they are good at

generating a high-quality single object. However, the models score low accuracies in the other skills, indicating that these skills (Count, Spatial, Scale, and Text Rendering) are still challenging with recent T2I models.

**Step-by-step generation improves challenging skills.** The bottom row of Table 1 shows that our VPGEN achieves high accuracy in Object skill and strongly outperforms other baselines in Count (+12.7% than Karlo), Spatial (+24.8% than Stable Diffusion v2.1) and Scale (+9.9% than Karlo) skills. This result demonstrates that our step-by-step generation method is effective in aligning image layout with text prompts, while also still having an interpretable generation program. All models achieve low scores on the Text Rendering skill, including our VPGEN. A potential reason why our VPGEN model does not improve Text Rendering score might be because the text-to-layout training datasets of VPGEN (Flickr30K Entities [17], COCO [18], and PaintSkills [19]) contain very few images/captions about text rendering, opening room for future improvements (*e.g.*, finetuning on a dataset with more images that focus on text rendering and including a text rendering module as part of our generation framework).

**Fine-grained analysis in Count/Spatial/Scale skills.** To better understand the high performance of our VPGEN in Count, Spatial, and Scale skills, we perform a detailed analysis of these skills with fine-grained splits. In Fig. 6, we compare our VPGEN (Vicuna+GLIGEN) and its closest baseline Stable Diffusion v1.4 on the three skills. **Overall**: VPGEN achieves better performance than Stable Diffusion on every split on all three skills. **Count**: While both models show difficulties with counting larger numbers, our VPGEN model achieves better performance with 50+ accuracy on all four numbers. **Spatial**: Our VPGEN model performs better on all six spatial relations. VPGEN shows higher accuracy on 2D relations (left/right/below/above) than on 3D depth relations (front/behind), while the Stable Diffusion is better at 3D relations than 2D relations. **Scale**: VPGEN generates more accurate layouts with two objects of different sizes (bigger/smaller), than layouts with two objects of similar sizes (same).

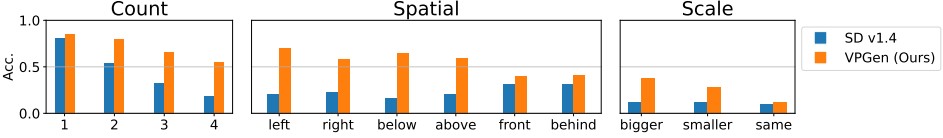

Figure 6: VPEVAL score comparison of our VPGEN (Vicuna+GLIGEN) and Stable Diffusion v1.4 on fine-grained splits in Count/Spatial/Scale skills.

### 5.3 Evaluation on Open-ended Prompts

Table 2 shows the VPEVAL score on the open-ended TIFA160 prompts. We calculate the score by averaging accuracy from the modules (see Fig. 5 for an example). The overall score trend of the baseline models (diffusion models > multimodal LMs) is consistent with the skill-based prompts. Unlike the trend in skill-based evaluation (Sec. 5.2), our VPGEN (Vicuna+GLIGEN) methods achieve similar performance to the Stable Diffusion baseline, while also providing interpretable generation steps. We perform an analysis of the open-ended prompts used for evaluation. In these prompts, object descriptions and attributes are the dominating prompt element (86.4% of elements), whereas descriptions of spatial layouts, only account for 13.6% of elements. See appendix for more

Table 2: VPEVAL scores on open-ended prompts (see Sec. 5.3). GLIGEN in VPGEN has Stable Diffusion v1.4 backbone. *F30: Flickr30K Entities, C: COCO, P: PaintSkills*.

| Model | Score (%) ↑ |
|---|---|
| Stable Diffusion v1.4 | 70.6 |
| Stable Diffusion v2.1 | 72.0 |
| Karlo | 70.0 |
| minDALL-E | 47.5 |
| DALL-E Mega | 67.2 |
| VPGEN (F30) | 71.0 |
| VPGEN (F30+C+P) | 68.3 |

details. For prompts from PaintSkills [19] where spatial layouts are more important, we find that our VPGEN scores much higher than Stable Diffusion v1.4 (71.0 *v.s.* 63.5) and Stable Diffusion v2.1 (71.0 *v.s.* 68.4). Note that GLIGEN used in VPGEN is based on Stable Diffusion v1.4, making it the closest baseline, and hence future versions of GLIGEN based on Stable Diffusion v2.1 or other stronger layout-to-image models will improve our results further. Also, we find that GLIGEN sometimes fails to properly generate images even when VPGEN generates the correct layouts (see following paragraph).

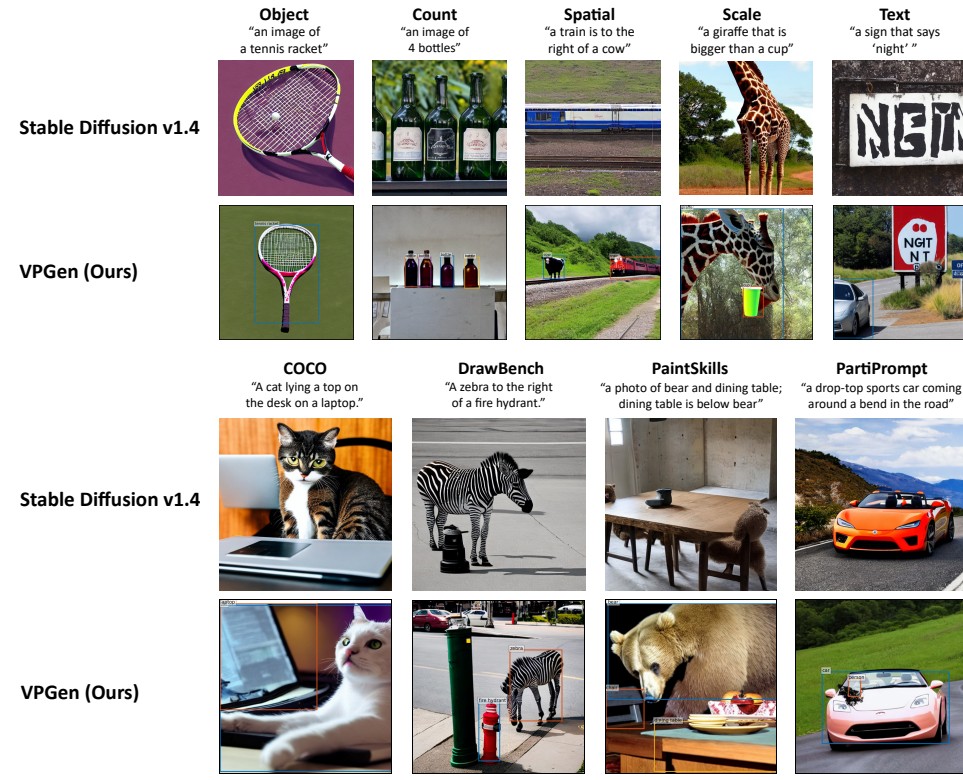

Figure 7: Images generated by Stable Diffusion v1.4 and our VPGEN (Vicuna 13B+GLIGEN) for skill-based (top) and open-ended (bottom) prompts.

**VPGEN sources of error: layout/image generation.** Since our VPGEN separates layout generation and image generation, we study the errors caused by each step. For this, we manually analyze the images generated from TIFA160 prompts, and label (1) whether the generated layouts align with the text prompts and (2) whether the final images align with the text prompts/layouts. GLIGEN sometimes fails to properly generate images even when Vicuna 13B generates the correct layouts, indicating that when better layout-to-image models become available, our VPGEN framework will achieve higher results. We include more details in the appendix.

**Qualitative Examples.** In Fig. 7, we provide example images generated by our VPGEN and Stable Diffusion v1.4 (the closest baseline) with various skill-based (top) and open-ended (bottom) prompts. See appendix for more qualitative examples with skill-based/open-ended prompts, prompts with unseen objects, counting $\geq 4$ objects, and error analysis.

## 5.4 Human Evaluation of VPEVAL

To measure alignments with human judgments and our VPEVAL, we compare the human-correlations of VPEVAL and other metrics on both skill-based prompts and open-ended prompts.

**Human correlation of VPEVAL on skill-based prompts.** We ask two expert annotators to evaluate 20 images, from each of the five baseline models, for each of the five skills with binary scoring (total $20 \times 5 \times 5 = 500$ images). The inter-annotator agreements were measured with Cohen's $\kappa$ [61] = 0.85 and Krippendorff's $\alpha$ [62] = 0.85, indicating 'near-perfect' ($\kappa > 0.8$ or $\alpha > 0.8$) agreement [63; 64; 62]. We compare our visual programs with captioning (with metrics BLEU [65], ROUGE [66], METEOR [67], and SPICE [68]), VQA, and CLIP (ViT-B/32) based evaluation. We use BLIP-2 Flan-T5 XL, the state-of-the-art public model for image captioning and VQA.

In Table 3, our VPEVAL shows a higher overall correlation with human judgments (66.6) than single module-based evaluations (CLIP, Captioning, and VQA). Regarding per-skill correlations, VPEVAL shows especially strong human correlations in Count and Text Rendering. As the BLIP-2 VQA module also shows strong correlation on Object/Spatial/Scale, we also experiment with VPEVAL$^{\dagger}$: using BLIP-2 VQA for `objectEval/spatialEval/scaleEval` modules (instead of Grounding

Table 3: Human correlation study of skill-based evaluation. We measure Spearman's $\rho$ correlation between human judgment and different automated metrics on the skill-based prompts (Sec. 4.3). VPEVAL†: using BLIP-2 VQA for `objectEval/spatialEval/scaleEval` modules.

| Eval Metric | Human-metric correlation (Spearman's $\rho$) ↑ | | | | | |
|---|---|---|---|---|---|---|
| | Object | Count | Spatial | Scale | Text | Overall |
| CLIP Cosine similarity (ViT-B/32) | 35.2 | 38.6 | 35.4 | 13.7 | 40.0 | 20.4 |
| BLIP-2 Captioning - BLEU | 11.9 | 31.4 | 26.3 | 24.0 | 23.6 | -3.4 |
| BLIP-2 Captioning - ROUGE | 15.7 | 26.5 | 28.0 | 12.2 | 28.3 | 11.9 |
| BLIP-2 Captioning - METEOR | 33.7 | 20.7 | 40.5 | 25.1 | 26.6 | 29.3 |
| BLIP-2 Captioning - SPICE | 56.1 | 20.9 | 40.6 | 27.3 | 18.6 | 28.1 |
| BLIP-2 VQA | **63.7** | 63.1 | 38.9 | 26.1 | 31.3 | 65.0 |
| VPEVAL | 34.5 | **63.8** | 48.9 | 29.4 | **85.7** | 73.5 |
| VPEVAL† | **63.7** | **63.8** | **51.2** | **29.5** | **85.7** | **79.0** |

DINO and DPT), which increases human correlation scores. Note that our object detection modules visually explain the evaluation results, as shown in Fig. 3, and we can have an even higher correlation when we have access to stronger future object detection models.

**Human correlation of VPEVAL on open-ended prompts.** We generate visual programs with our program generation LM on TIFA160 prompts [36]. The dataset consists of 800 images (160 prompts × 5 T2I models) and human judgments (1-5 Likert scale) along with other automatic metrics (BLIP-2 captioning, CLIP cosine similarity, and TIFA with BLIP-2) on the images. Table 4 shows that our VPEVAL achieves a better human correlation with TIFA (BLIP-2), and our VPEVAL† version achieves an even higher correlation. The results indicate that using various interpretable modules specialized in different skills complements each other and improves human correlation, while also providing visual+textual explanations.

**Human analysis on the generated programs.** Lastly, we also measure the faithfulness of the generated evaluation programs. For this, we sample TIFA160 prompts and analyze the evaluation programs by (1) how well the modules cover elements in the prompt; (2) how accurate the module outputs are when run. We find that programs generated by our VPEVAL have very high coverage over the prompt elements and high per-module accuracy compared to human judgment. We include more details in the appendix.

Table 4: Human correlation on open-ended evaluation with Spearman's $\rho$.

| Metrics | $\rho$ (↑) |
|---|---|
| *BLIP-2 Captioning* | |
| BLEU-4 | 18.3 |
| ROUGE-L | 32.9 |
| METEOR | 34.0 |
| SPICE | 32.8 |
| *Cosine-similarity* | |
| CLIP (ViT-B/32) | 33.2 |
| *LM + VQA module* | |
| TIFA (BLIP-2) | 55.9 |
| *LM + multiple modules* | |
| VPEVAL (Ours) | 56.9 |
| VPEVAL† (Ours) | **60.3** |

## 6 Conclusion

We propose two novel visual programming frameworks for interpretable/explainable T2I generation and evaluation: VPGEN and VPEVAL. VPGEN is a step-by-step T2I generation framework that decomposes the T2I task into three steps (object/count generation, layout generation, and image generation), leveraging an LLM for the first two steps, and layout-to-image generation models for the last image generation step. VPGEN generates images more accurately following the text descriptions (especially about object counts, spatial relations, and object sizes) than strong T2I baselines, while still having an interpretable generation program. VPEVAL is a T2I evaluation framework that uses interpretable evaluation programs with diverse visual modules that are experts in different skills to measure various T2I skills and provide visual+textual explanations of evaluation results. In our analysis, VPEVAL presents a higher correlation with human judgments than single model-based evaluations, on both skill-specific and open-ended prompts. We hope our work encourages future progress on interpretable/explainable generation and evaluation for T2I models.

**Limitations & Broader Impacts.** See Appendix for limitations and broader impacts discussion.

## Acknowledgement

We thank the reviewers for their valuable comments and feedback. This work was supported by ARO W911NF2110220, DARPA MCS N66001-19-2-4031, NSF-AI Engage Institute DRL211263, ONR N00014-23-1-2356, and DARPA ECOLE Program No. HR00112390060. The views, opinions, and/or findings contained in this article are those of the authors and not of the funding agency.

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

# Appendix

In this appendix, we provide qualitative examples (Appendix A), additional VPGEN analysis (Appendix B), VPGEN error analysis (Appendix C), VPEVAL error analysis (Appendix D), VP-GEN/VPEVAL implementation details (Appendix E), TIFA prompt element analysis details (Appendix F), limitations and broader impacts (Appendix G), and license information (Appendix H).

## A  Qualitative Examples

In Fig. 8 and Fig. 9, we provide example images generated by our VPGEN and Stable Diffusion v1.4 (the closest baseline of VPGEN) with various skill-based and open-ended prompts, respectively. In Fig. 10, we show VPGEN generation examples of placing objects that are unseen during the text-to-layout finetuning of Vicuna 13B.

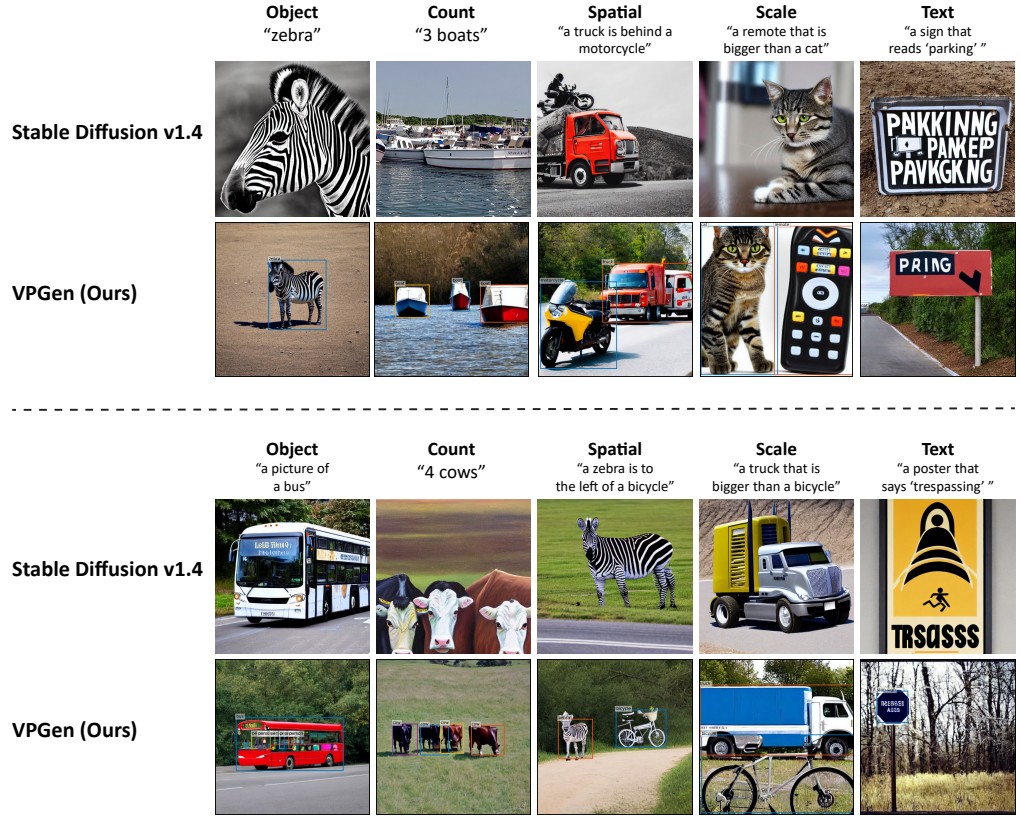

Figure 8:  Images generated by Stable Diffusion v1.4 and our VPGEN (Vicuna 13B+GLIGEN) for our skill-based prompts.

## B  Additional VPGEN Analysis

In the following, we provide additional analysis of VPGEN, including counting prompts, bounding box quantization, using GPT-3.5-Turbo as layout generation LM, and visual quality metric (FID).

**Counting prompts: More than 4 objects & Unspecified number of objects.**   In addition to the Count skill results with counts of 1 to 4 in the main paper Fig. 6, we provide additional results of VPGEN with counts of up to 7 in Table 5 and show generation examples in Fig. 11 (a) - (d). As expected, the trend continues from Fig. 6, where higher counts are more challenging. We also experiment with prompts without specific numbers. We find that VPGEN can successfully generate layouts/images even when the prompts do not have explicit counts. As shown in Fig. 11 (e), our VPGEN generates two Pikachus from the prompt "Pikachus on the table".

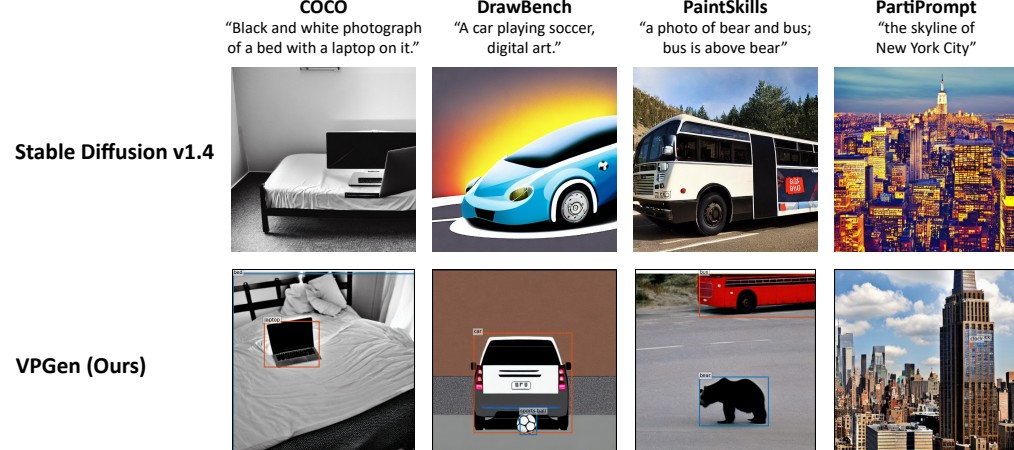

Figure 9: Images generated by Stable Diffusion v1.4 and our VPGEN (Vicuna 13B+GLIGEN) for open-ended prompts.

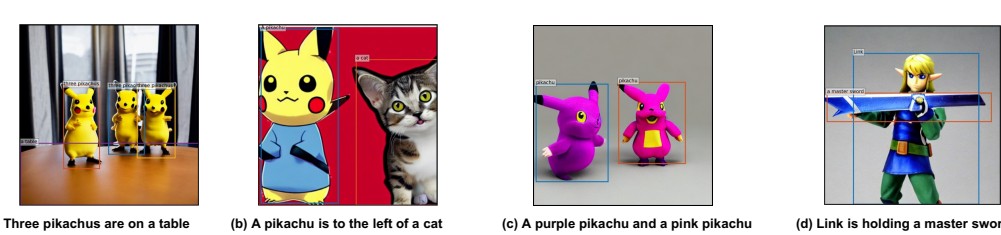

Figure 10: Images generated by VPGEN (Vicuna 13B+GLIGEN) from prompts including objects (*e.g.*, 'Pikachu', 'Link', 'master sword') that are not included in text-to-layout annotations (Flickr30K Entities, COCO, PaintSkills) for Vicuna fine-tuning.

**Bounding box quantization.** As mentioned in the main paper, we use normalized and quantized coordinates with 100 bins to represent bounding boxes, following LM-based object detection works [49; 50]. In Table 6, we compare 100 and 1000 bins for the bounding boxes. Increasing the granularity of the bounding box to 1000 does not increase the accuracy in both skill-based and open-ended prompts, suggesting that the current 100-bin quantization does not hurt the accuracy of object placements.

**GPT-3.5-Turbo *v.s.* Vicuna 13B.** In our initial experiment, we tested GPT-3.5-Turbo to generate spatial layouts by showing in-context examples and found that the generated layouts were often inaccurate or not meaningful. For example, GPT-3.5-Turbo often generates a list of bounding boxes that is the same size as the entire image (*e.g.*, `[object 1 (0,0,99,99), object 2 (0,0,99,99) object3 (0,0,99,99)]`). We conjecture that this is because their training corpus might not include many bounding boxes. Then, we collected text-layout annotations and trained a language model to generate spatial layouts from text prompts.

For a quantitative comparison, we implement VPGEN with GPT-3.5-turbo with 36 in-context examples that cover different skills and compare it with Vicuna 13B trained on Flickr30k+COCO+PaintSkills. As shown in Table 7, Vicuna 13B based VPGEN shows higher skill-based and open-ended VPEval accuracies than GPT-3.5-Turbo based VPGEN.

**Visual quality metric (FID).** In Table 8, we compare our VPGEN (Vicuna13B + GLIGEN) to its backbone Stable Diffusion v1.4, in FID [30] (30K images of COCO val 2014 split) and VPEval Accuracy. Both VPGEN checkpoints show better skill-based accuracy than Stable Diffusion v1.4 while achieving comparable open-ended accuracy. and FID. In FID (lower the better), we find VPGEN (Flickr30k) < Stable Diffusion v1.4 < VPGEN (Flickr30k+COCO+Paintskills). We think that a bit of increase (but still reasonably good) in the FID of the Flickr30k+COCO+Paintskills

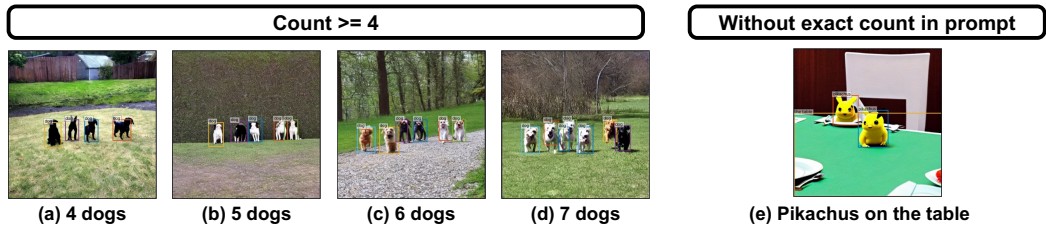

| Count >= 4 | | | | Without exact count in prompt |
|---|---|---|---|---|
| (a) 4 dogs | (b) 5 dogs | (c) 6 dogs | (d) 7 dogs | (e) Pikachus on the table |

Figure 11: Images generated by our VPGEN with prompts requiring counting skills. (a) - (d) show the generation examples with count $\geq 4$. (e) shows generation from a prompt that does include exact count ('pikachus').

Table 5: VPEVAL Count skill accuracy of VPGEN with different number of objects.

|  | VPEval Count Acc. (%) ↑ | | | | | | |
|---|---|---|---|---|---|---|---|
|  | 1 | 2 | 3 | 4 | 5 | 6 | 7 |
| VPGEN | 85.3 | 80.1 | 65.6 | 55.2 | 48.0 | 35.3 | 28.1 |

checkpoint is because the layouts of PaintSkills are different from those of natural scenes (COCO and Flickr30k).

**Training with real *v.s.* synthetic images.** Our layout generation module (Vicuna 13B) in VP-GEN is trained on a mix of both 3D simulator-based synthetic (PaintSkills) and real-world images (MSCOCO/Flickr30K). We conduct an ablation study of training only with real *v.s.* synthetic images. We train the Vicuna on only PaintSkills and only Flickr30K and compare their results on both skill-based and open-ended prompts in Table 9. For skill-based prompts, PaintSkills training shows slightly higher average accuracy (46.0) than Flickr30K training (43.9). For open-ended prompts, PaintSkills training gets a VPEval score of (65.6), lower than Flickr30K training (71.0). This indicates that training on synthetic data can inject knowledge of specific skills such as counting and spatial relation understanding, but it is hard to cover diverse objects and attributes only with synthetic data.

## C    VPGEN Error Analysis

**Layout *v.s.* Image generation on skill-based prompts.** We analyze the performance of each step in the VPGEN pipeline to determine how much error is propagated with the following four metrics.

1. Object Recall: The ratio of whether correct objects are included in the generated layouts.
2. Object Count: The ratio of the correct number of objects is included in the generated layouts.
3. Layout Accuracy: VPEval accuracy of the generated layouts (using the generated layouts as the object detection results).
4. Image Accuracy: VPEval accuracy of the final image by GLIGEN (in the main Table 1).

As shown in Table 10, the layout accuracy is much higher than image accuracy, especially for Count/Spatial skills. This indicates that the major error source is the image rendering step (with GLIGEN), and using a more accurate layout-to-image generation model in the future would improve the accuracy of our VPGEN pipeline.

**Human evaluation on open-ended prompts.** As mentioned in the main paper Sec. 5.3, we ask two expert human annotators to evaluate 50 randomly sampled prompts from TIFA160 [36], with the following two metrics: (1) layout accuracy: if the generated layouts align with the text prompts; (2) image accuracy: if the generated images align with the text prompts. The human annotators show high inter-annotator agreements for both layout accuracy (Cohen's $\kappa = 0.73$, Krippendorff's $\alpha = 0.73$) and image accuracy (Cohen's $\kappa = 0.87$, Krippendorff's $\alpha = 0.87$). The Vicuna 13B's layout accuracy is (92%) higher than GLIGEN's image correctness (65%). The result is consistent with the skill-based prompts error analysis result (Table 10) and suggests that a better layout-to-image generation model could further improve our VPGEN framework.

Table 6: Bounding box quantization: 1000 *v.s.* 100 bins.

| # Bins | VPEval skill-based Acc. (%) ↑ | VPEval open-ended Acc. (%) ↑ |
| --- | --- | --- |
| 1000 | 49.8 | 68.2 |
| 100 (default) | **51.0** | **68.3** |

Table 7: GPT-3.5-Turbo *v.s.* Vicuna 13B in VPGEN.

| Model | VPEval skill-based Acc. (%) ↑ | VPEval open-ended Acc. (%) ↑ |
| --- | --- | --- |
| GPT-3.5-Turbo (36 examples) + GLIGEN | 40.7 | 65.5 |
| Vicuna 13B + GLIGEN | **51.0** | **68.3** |

**Common error categories.**   We show some common categories of errors from the text-to-layout generation step (with Vicuna 13B) and from the layout-to-image generation step (with GLIGEN), respectively. As shown in Fig. 12, the Vicuna 13B sometimes (a) generates objects that are not specified in prompts (*e.g.*, 'chair' in the 1st example) or (b) misses objects that are mentioned in prompts (*e.g.*, 'building' in the 5th example). As shown in Fig. 13, the GLIGEN sometimes (a) generates wrong objects (*e.g.*, 'suitcase', instead of 'bench', in the 1st example), (b) misses some objects (*e.g.*, 'umbrella' in the 3rd example), or (c) misses object details (*e.g.*, 'closed oven' instead of 'oven with the door open' in the 5th example).

**Challenging, non-canonical prompts.**   In Fig. 14, we show generation examples with challenging, non-canonical prompts (a) "A realistic photo of a Pomeranian dressed up like a 1980s professional wrestler with neon green and neon orange face paint and bright green wrestling tights with bright orange boots" (from DrawBench [24]) and (b) "a circle with 3 cm radius and a circle with 6 cm radius". Our VPGEN understands the important parts of the prompts (*e.g.*, generating 'realistic', 'Pomeranian dog', 'bright orange boots' and generating layouts of two circles in different sizes), but misses some aspects (*e.g.*, the bigger circle is not twice the size of the smaller circle). This is probably because the training prompts (COCO, PaintSkills, Flickr30k Entities) do not include many prompts written in these styles. We believe that scaling datasets with various sources can further improve VPGEN.

## D   VPEVAL Error Analysis

**Evaluation program analysis: coverage and accuracy.**   For the evaluation program analysis (main paper Sec. 5.4), we experiment with two expert annotators using 50 randomly sampled TIFA160. We ask two expert annotators to analyze the evaluation programs in two criteria: (1) how well the evaluation program covers the content from the text prompt; (2) how accurately each evaluation module evaluates the images. To make it easier for the annotators, for (2) module accuracy analysis, we only ask them to check the output of three randomly sampled evaluation modules. We find that, on average, our evaluation programs cover 94% of the prompt elements, and our modules are accurate 83% of the time. This shows that our VPEVAL evaluation programs are effective in selecting what to evaluate and are highly accurate while also providing interpretability and explainability. In Fig. 15, we visualize our evaluation programs that cover different parts of the prompts and evaluation modules that provide accurate outputs and visual+textual explanations.

## E   VPGEN/VPEVAL Implementation Details

### E.1   VPGEN Details

**Training layout-aware LM.** To obtain layout-aware LM, we use Vicuna 13B [16], a public state-of-the-art chatbot language model finetuned from LLaMA [51]. We use parameter-efficient finetuning with LoRA [52] to preserve the original knowledge of LM and save memory during training and inference. We collect the text-layout pair annotations from training sets of three public datasets: Flickr30K entities [17], MS COCO instances 2014 [18], and PaintSkills [19], totaling 1.2M examples. We use randomly selected 2,000 examples for the validation set and use the rest for training. For the object/count generation step, we set the maximum counts for a single object class as 7. We train

Table 8: VPEval accuracy and FID of VPGen (Vicuna 13B + GLIGEN) and its closest baseline Stable Diffusion v1.4. *F30: Flickr30K Entities, C: COCO, P: PaintSkills*.

| Model | VPEval Skill-based Acc. (%) ↑ | VPEval Open-ended Acc. (%) ↑ | FID (COCO 30K) ↓ |
|---|---|---|---|
| Stable Diffusion v1.4 | 37.7 | 70.6 | 16.5 |
| VPGEN (F30) | 43.9 | **71.0** | **15.9** |
| VPGEN (F30+C+P) | **51.0** | 68.3 | 20.1 |

Table 9: Training Vicuna 13B with real *v.s.* synthetic data. *F30: Flickr30K Entities, P: PaintSkills*.

| Model | VPEval skill-based Acc. (%) ↑ | VPEval open-ended Acc. (%) ↑ |
|---|---|---|
| VPGEN (F30; real) | 43.9 | **71.0** |
| VPGEN (P; synthetic) | **46.0** | 65.6 |

Vicuna 13B with per-gpu batch size 96 (= 24 batch x 4 gradient accumulation). When training Vicuna 13B with Flickr30K+COCO+PaintSkills dataset, we train the model for 2 epochs. When training only with Flickr30K dataset, we train the model for 6 epochs, to rough match the training time. Training takes 26 hours with 4 A6000 GPUs (each 48GB).

**GLIGEN inference details.** We use Huggingface Diffusers [69] implementation of GLI-GEN [20][2]. Following the default configuration, we use `gligen_scheduled_sampling_beta` = 0.3, `num_inference_steps` = 50, and fp16 precision during inference.

**Inference time.** On a single A6000 GPU, VPGEN takes around 6s (2s for Vicuna 13B and 4s for GLIGEN) to generate an image.

## E.2  VPEVAL Details

**VPEVAL prompts and evaluation programs.** Our skill-based benchmark has an almost uniform distribution of objects and relations among the text prompts. This ensures that generation models cannot achieve high scores by performing highly on a few common objects, relations, and counts, *etc*. We create skill-specific prompts by composing templates with lists of objects (from 80 COCO objects), counts (1-4), relations (2D: above, below, left, right; 3D: front, behind), scale (smaller, bigger, same), and text (31 unique words). Our evaluation can be extended to any number of objects/counts/relations/*etc*. In total, our skill-based benchmark has 400/1000/1000/1000/403 prompts for the object/count/spatial/size/text skills, respectively. In Table 11, we show templates and evaluation programs used for each skill.

**Captioning/VQA/CLIP baseline details.** For captioning, we generate a caption with BLIP-2 [56] and calculate text similarity metrics between the caption and the original text prompt. For VQA, we give the BLIP-2 (Flan-T5 XL) the image, a question, and a yes/no answer choice, with a prompt template ``Question: {question} Choices: yes, no Answer:''. The {question} is a formatted version of the text prompt (*e.g.*, "a photo of a dog" → "is there a dog in the image?"). For the spatial and scale skill, we ask three questions: if object A is present (*e.g.*, "is there a dog in the image?") if object B is present (*e.g.*, "is there a cat in the image?"), and if the relationship between them is true (*e.g.*, "is the cat behind the dog?"). We mark each image as correct if the VQA model outputs 'yes' for all three questions. The first two questions prevent accidental false positives if an

---

[2]https://github.com/gligen/diffusers

Table 10: Step-wise error analysis of VPGen (Vicuna+GLIGEN). For the Spatial skill, 'front' and 'behind' splits are skipped in this table since our Vicuna 13B does not generate depth information.

| Skills | Vicuna 13B | | | GLIGEN |
|---|---|---|---|---|
| | Object Recall (%) ↑ | Object Count (%) ↑ | Layout Accuracy (%) ↑ | Image Accuracy (%) ↑ |
| Object | 99.2 | 98.8 | 99.2 | 96.8 |
| Count | 99.1 | 98.8 | 98.8 | 72.2 |
| Spatial | 98.3 | 98.2 | 87.5 | 63.3 |
| Scale | 92.8 | 92.6 | 38.2 | 26.3 |

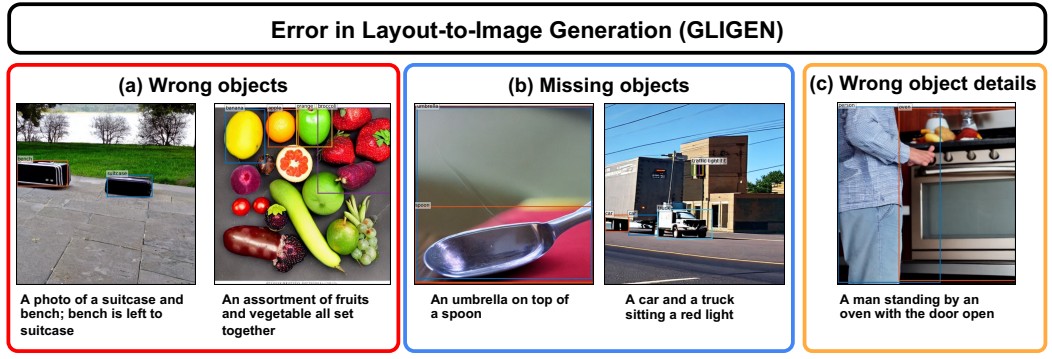

Figure 12: Text-to-Layout generation error examples. The Vicuna 13B sometimes (a) generates objects that are not specified in prompts (*e.g.*, 'chair' in the 1st example) or (b) misses objects that are mentioned in prompts (*e.g.*, 'building' in the 5th example).

**Error in Layout-to-Image Generation (GLIGEN)**

**(a) Wrong objects**

A photo of a suitcase and bench; bench is left to suitcase

An assortment of fruits and vegetable all set together

**(b) Missing objects**

An umbrella on top of a spoon

A car and a truck sitting a red light

**(c) Wrong object details**

A man standing by an oven with the door open

Figure 13: Layout-to-Image generation error examples. The GLIGEN sometimes (a) generates wrong objects (*e.g.*, 'suitcase', instead of 'bench', in the 1st example), (b) misses some objects (*e.g.*, 'umbrella' in the 3rd example), or (c) misses object details (*e.g.*, 'closed oven' instead of 'oven with the door open' in the 5th example).

object is missing. For CLIP (ViT-B/32) [32], we encode the generated image and text prompts, and take the cosine similarity between the visual and text features as the final score.

**Inference time.** On a single A6000 GPU, each VPEVAL module typically takes less than 1s to run.

**Program generation API cost.** Using GPT-3.5 Turbo[3] as the program generator, VPEVAL takes about $0.007 (less than 1 cent) to generate programs from a prompt (4K input tokens, 0.5K output tokens).

## F TIFA Prompt Element Analysis Details

To better understand the characteristics of the TIFA prompts [36] that we use in open-ended evaluation, we randomly sample 20 prompts and label whether each prompt has any elements that require an understanding of spatial layouts, as mentioned in the main paper Sec. 5.3. Each prompt has multiple elements about objects, attributes, spatial layouts, *etc*. For example, "a zebra to the right of a fire hydrant." has the elements: 'zebra', 'right', and 'fire hydrant'. From the 20 prompts, there are 81 total elements, and only 11/81 (13.6%) are relevant to object layouts (*e.g.*, 'right'). The remaining 70 elements are all related to objects or attributes (*e.g.*, "a white cat with black ears and markings" has the elements 'cat', 'white cat', 'ears', 'black ears', 'markings', and 'black markings').

---

[3]https://openai.com/pricing

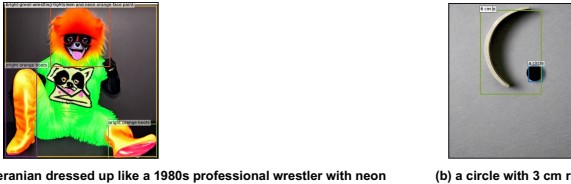

(a) A realistic photo of a Pomeranian dressed up like a 1980s professional wrestler with neon green and neon orange face paint and bright green wrestling tights with bright orange boots

(b) a circle with 3 cm radius and a circle with 6 cm radius

Figure 14: Images generated by VPGEN (Vicuna+GLIGEN) from challenging, non-canonical prompts.

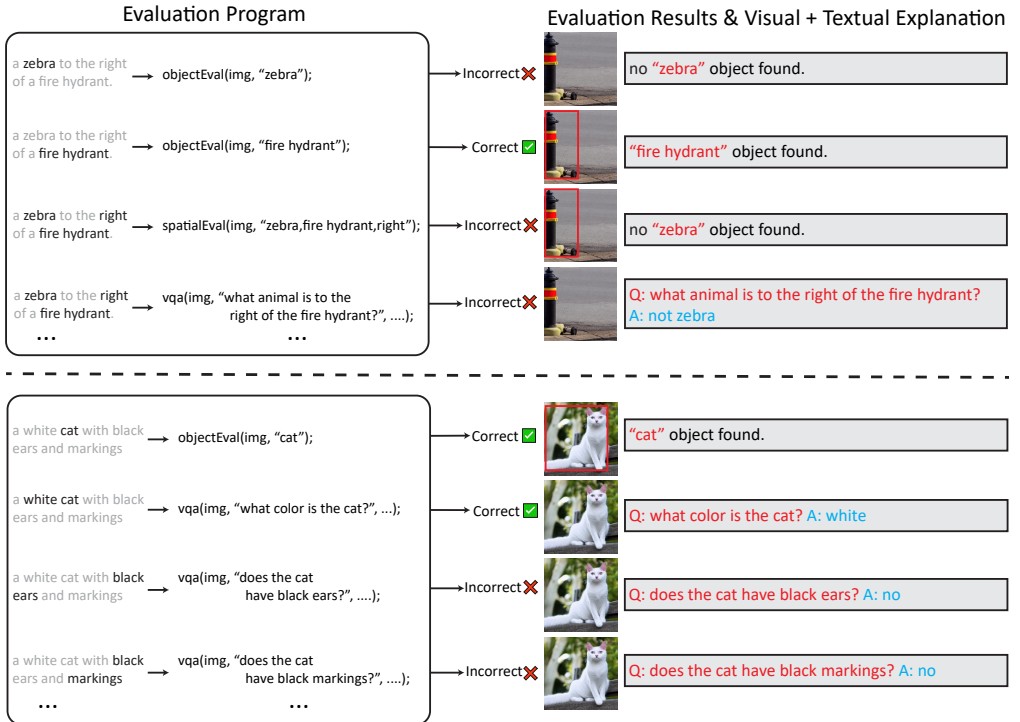

Figure 15: Examples of our VPEVAL evaluation programs and module outputs. Each module covers different parts of the text prompts (see Appendix D for discussion).

# G   Limitation and Broader Impacts

**Limitations.**   Since our LMs were primarily trained with English-heavy datasets, they might not well understand prompts written in non-English languages. Likewise, as our generation/evaluation modules were primarily trained with natural images, they might not well handle images in different domains (*e.g.*, fine-arts, medical images, *etc*). Note that our framework supports easy extensions so that users can update existing modules or add new ones with better domain-specific knowledge.

Generating evaluation programs using LLMs can be costly both in terms of computational resources and price (API cost). Thus, we release the evaluation programs so that users do not have to re-generate the evaluation programs. In addition, we will release a public LM (finetuned for evaluation program generation using ChatGPT outputs) that can run on local machines.

**Broader Impacts.**   While our interpretable VPGEN framework can be beneficial to many applications (*e.g.*, user-controlled image generation/manipulation and data augmentation), it could also be used for potentially harmful cases (*e.g.*, creating false information or misleading images), like other image generation frameworks.

Table 11: The prompt templates and evaluation programs for skill-based evaluation. We also insert grammar tokens like `<a>` that are replaced with "an" or "a" to maintain correct grammar.

| Skill | Object | Count | Spatial | Scale | Text Rendering |
|---|---|---|---|---|---|
| Prompt | `<objA>`
`<a> <objA>`
a photo of `<a> <objA>`
an image of `<a> <objA>`
a picture of `<a> <objA>` | `<N> <objA>`
a photo of `<N> <objA>`
a picture of `<N> <objA>`
an image of `<N> <objA>`
`<N_EN> <objA>`
a photo of `<N_EN> <objA>`
a picture of `<N_EN> <objA>`
an image of `<N_EN> <objA>` | `<a2> <objB>` is `<tothe>`
`<rel><of> <a1> <objA>` | `<a2> <objB>` that is
`<scale>` than `<a1> <objA>` | a sign that reads '`<text>`'
a book cover that reads '`<text>`'
a poster that reads '`<text>`'
a sign that says '`<text>`'
a book cover that says '`<text>`'
a poster that says '`<text>`'
a storefront with '`<text>`' written on it
a storefront with '`<text>`' written
a storefront with '`<text>`' displayed
a piece of paper that says '`<text>`'
a piece of paper that reads '`<text>`'
a piece of paper that says '`<text>`' on it
a piece of paper that reads '`<text>`' on it |
| Evaluation Program | objectEval(img, `<objA>`) | countEval(img, `<objA>`, `<N>`) | spatialEval(img, `<objA>`,`<objB>`,`<rel>`) | scaleEval(img, `<objA>`,`<objB>`,`<scale>`) | textEval(img, `<text>`) |

# H    License

We will make our code and models publicly accessible. We use standard licenses from the community and provide the following links to the licenses for the datasets, codes, and models that we used in this paper. For further information, please refer to the links.

**FastChat (Vicuna):** Apache License 2.0

**GLIGEN:** MIT

**Grounding DINO:** Apache License 2.0

**DPT:** MIT

**LAVIS (BLIP-2):** BSD 3-Clause

**TIFA:** Apache License 2.0

**minDALL-E:** Apache License 2.0, CC-BY-NC-SA 4.0

**DALL-E mini (DALL-E Mega):** Apache License 2.0, MIT

**EasyOCR:** Apache License 2.0

**OpenAI API (ChatGPT):** MIT

**Diffusers:** Apache License 2.0

**Transformers:** Apache License 2.0

**PyTorch:** BSD-style

