# OpenReview forum: "Visual Programming for Step-by-Step Text-to-Image Generation and Evaluation"
_NeurIPS.cc/2023/Conference — NeurIPS 2023 poster_

### Official Review · Reviewer_bJid · 2023-07-03

**Soundness:** 3 good
**Presentation:** 3 good
**Contribution:** 3 good
**Rating:** 6
**Confidence:** 3

**Summary:**

This paper makes two contributions. VPGen is a T2I generation framework that first generates object/count, then layout, and finally image. VPEval is a T2I evaluation pipeline that provides a more comprehensive analysis correlated to human. Authors demonstrate that the step-by-step VPGen approach generates images more aligned with the input text compared to stable diffusion and other baselines. This is evaluated using the newly proposed VPEval scores.

**Strengths:**

The proposed VPGen decomposition demonstrates better controllability for image generation, especially in terms of following the exact object count and spatial relation from text prompts. The new VPEval score offers a more comprehensive evaluation beyond just image quality. I believe this score will be useful for evaluating controllable generation.

**Weaknesses:**

Result section lacks the standard evaluation metric for image generation such as the FID and IS score. While the focus of this paper is on better controllability, and the proposed VPEval is clearly a better choice, I still like to see some benchmark on the image quality and diversity.

First step of VPGen seems limited. In the examples provided, text has to include the exact number for each object. So is the first step just doing text parsing? Will it still work if I give it a sentence like “a small cozy office room”?

Also the motivation for training a LM is not clear to me. Many existing works can generate the layout bounding boxes directly, what is the extra benefit of training a LM? Can you do interactive editing?

**Questions:**

(1) It will be great if authors can also provide the standard evaluation benchmarks.

(2) Can authors explain their design choice for using LM in VPGen?

(3) More visualizations results will be appreciated.

**Limitations:**

Yes, authors adequately addressed the limitations.

---

> ### Author Rebuttal · Authors · 2023-08-09
>
> Thank you for the useful feedback and for pointing out our strengths in providing useful image generation and evaluation frameworks. We hope to address your questions and concerns below.
>
> **W1. Standard image quality metric (FID).** Please see the general response.
>
> **W2-1. Prompts without an exact number of counts.** Thanks for the suggestion, users can indeed provide prompts without specific numbers. We find that VPGen can successfully generate layouts/images even when prompts do not have explicit counts. As shown in Figure B (e) in the attached PDF, our VPGen generates two Pikachus from the prompt “Pikachus on the table”.
>
>
> **W2-2. Prompts where details are unspecified.**
> Regarding your specific prompt “a small cozy office room”, we show two images from two different checkpoints (Flickr30k vs. Flickr30k+COCO+Paintskills) in Figure C (c) in the attached PDF. Note that the bounding boxes of the Flickr30k entities dataset are explicitly grounded to the Flickr30k captions, while the COCO captions and bounding boxes are annotated separately. This results in two different layout generation styles: 1) COCO-style - LM generates object layouts even if they are not explicitly mentioned in the prompt; 2) Flickr30k entities style - LM only generates objects that are explicitly mentioned in the prompt, and the downstream image generation model (GLIGEN) generates the image details. Users can construct their text-layout grounding style for their use cases. We add visualizations of such prompts to the paper.
>
>
> **W3 & Q2. Motivation for training an LM for VPGen.** In our initial experiment, we tested ChatGPT to generate spatial layouts by showing in-context examples and found the generated layouts were often inaccurate or not meaningful. For example, ChatGPT often generates a list of bounding boxes that is the same size as the entire image (e.g., `[object 1 (0,0,99,99), object 2 (0,0,99,99) object3 (0,0,99,99)]`). We conjecture that this is because their training corpus might not include a lot of bounding boxes. Then we decided to collect text-layout annotations and finetune a language model to generate spatial layouts from text prompts.
>
> For a quantitative comparison, we implement VPGen with ChatGPT-3.5-turbo with 36 in-context examples covering different skills and compare it with (open-sourced) Vicuna13B trained on Flickr30k+COCO+PaintSkills. As shown in Table B, Vicuna13B-based VPGen shows higher skill-based and open-ended VPEval accuracies than ChatGPT-based VPGen. We will add this interesting result to the paper.
>
> Table B. ChatGPT vs. Vicuna13B.
>
>
> | Model  | VPEval Skill-based Acc. (%) | VPEval Open-ended Acc. (%) |
> |-----------|------------|-------------|
> | ChatGPT (36 examples) + GLIGEN |   42.2 |  66.3 |
> | Vicuna13B + GLIGEN | **47.0**  | **70.0**  |
>
>
>
> **Q3. More Visualizations.** Thanks for the suggestion. In the attached PDF, we provide different visualization results covering unseen objects (Figure A), counting (Figure B), challenging/complex prompts (Figure C), and layout error categorization (Figure D). We will include these and more visualizations in the paper.

---

> > ### Comment · Reviewer_bJid · 2023-08-18
> >
> > I thank authors for the detailed response and really appreciate their effort. Authors clarified many points and added crucial experiments in their rebuttal. Most of my concerns have been addressed. One comment is that I agree with other reviewers the evaluation pipeline might not be robust in some cases. I do encourage authors to list out the possible failure cases in their updated version. The overall quality has improved and I raised my score accordingly.

---

> > > ### Author Response · Authors · 2023-08-18
> > >
> > > Thanks for your supportive and positive discussion! We are glad that you appreciated our clarifications and additional experiments.
> > >
> > > Yes, following your suggestion, we will definitely include more qualitative examples in the final version.

---

### Official Review · Reviewer_PtVL · 2023-07-04

**Soundness:** 3 good
**Presentation:** 3 good
**Contribution:** 3 good
**Rating:** 7
**Confidence:** 3

**Summary:**

This paper proposes a visual-program-based evaluation method VPEval to evaluate text-to-image models. Their method relies on LLM which can call different expert models in different tasks like object detection, OCR, spatial understanding, etc. to evaluate the consistency between the text and the generated image. The human evaluation shows that compared to traditional metrics, their evaluation method aligns with human better. Besides, they introduce a novel interpretable step-by-step text-to-image generation framework VPGen to improve the consistency between generated images and texts in count, spatial, and scale skills.

**Strengths:**

1.	The paper writing is clear
2.	Their framework VPGen is simple and novel. I like the idea to first train a LLM that generates image high-level layouts from the given prompt, then generates the image based on the layout.
3.	VPGen is effective with improved consistency in counting/spatial/scale relationships.
4.	The evaluation framework VPEval based on visual program is also novel and has better interpretability compared to previous methods. Besides, it aligns with human judgement better.

**Weaknesses:**

1.	This paper is highly related and based on visual programing. Although the author cites the related papers like Visual Programming [11] and ViperGPT [12] in the related work section L110, a brief introduction of visual programming and a discussion of the difference between [11,12] and this paper are missing.

**Questions:**

1.	As the evaluation is based on ChatGPT, it would be good to include a API cost analysis.

**Limitations:**

The authors listed and discussed the limitations including the suboptimal performance with non-English prompts and unnatural images. Potential broader impacts like generating harmful images are also discussed.

---

> ### Author Rebuttal · Authors · 2023-08-09
>
> Thank you for the useful feedback and for pointing out our strengths of the effectiveness of VPGen and VPEval, as well as clear writing. We hope to address your questions and concerns below.
>
> **W1. Introduction of Visual Programming.** We’d like to bring your attention to Sec 1, where we explain how the recent LLM+visual modules (e.g., Visual Programming and ViperGPT) work (L38-31; “LLMs can also tackle certain vision-and-language tasks such as visual question answering and visual grounding, by generating visual programs (VP) that can control external visual modules and combine their outputs to get the final response”), and how our frameworks are different from them (L31-34; “However, no prior works have shown an analysis of combining LLMs and different visual modules for the challenging text-to-image (T2I) generation task. Our work proposes two novel interpretable/explainable VP frameworks combining LLMs and visual modules for T2I generation and evaluation”). In the next version, we will further expand the introduction of the Visual Programming framework and discuss the difference between the existing works and ours to make it more clear.
>
> **Q1. VPEval API Costs.** Thanks for the suggestion. VPEval costs about $0.007 (less than 1 cent) to evaluate one image with the ChatGPT 3.5 turbo API. We will add this information to the paper. Please also note that we will release all the generated programs from our evaluations for others to use (L225-227).

---

> > ### Comment · Reviewer_PtVL · 2023-08-18
> >
> > Thanks for the response. I appreciate the additional qualitative examples and quantitative evaluation. Given the good novelty and the simplicity of the methodology, I'll increase my score to 7.

---

> > > ### Author Response · Authors · 2023-08-18
> > >
> > > Thanks for your supportive comments! We are glad that you appreciated our additional examples + evaluation (and novelty + simplicity of our methodology).
> > >
> > > BTW, just a gentle note - the increase in the score (as you mentioned in your last comment) may not have gotten saved in the system.
> > >
> > > Thanks again!

---

### Official Review · Reviewer_GZ2e · 2023-07-09

**Soundness:** 4 excellent
**Presentation:** 4 excellent
**Contribution:** 4 excellent
**Rating:** 8
**Confidence:** 4

**Summary:**

This paper extends previous work in the vision and language space that use visual programs as an intermediate step, to the problem of text to image generation and subsequently, its evaluation. It proposes VPGen, a neuro-symbolic method that is composed of specific modules that count objects, generates layouts using a LLM and then generates an image using GLIGEN, a layout-to-image model. This allows for interpretable inspection of the intermediate steps as well as allows for more controllability in the image generation process. The paper also proposes VPEval, a method to evaluate T2I models using specific modules that also generate interpretable programs inspecting specific skills such as counting, existence of object, spatial, scale and text rendering. The proposed method (VPGen) outperforms previous approaches on skills such as count, spatial and scale, demonstrating that the approach allows for better fine-grained control over the image generation process.

**Strengths:**

## Quality, Originality and Significance
* There have been several methods that were proposed over the last year for text to image generation, such as stable diffusion, Imagen, Parti, etc but the community still lacked an adequate method for evaluating the quality of the generated image in a way that is also interpretable. This paper provides both, an interpretable text to image generation pipeline that either outperforms or is competitive with existing (open source) state of the art text to image generation models, as well as a way to evaluate them.
* Improvements in layout generation by using an LM, allowing generalization to unseen concepts instead of depending on pre-defined class set broadens the scope of the type of prompts that can be incorporated into this interpretable pipeline.
* Human judgements and correlation analysis of existing evaluation methods compared with the proposed method demonstrates that VPEval is a suitable metric for T2I generation both in terms of interpretability while also being correlated with human judgement.

## Clarity
* The paper is very well written, and is very easy to follow.
* The paper together with the appendix provides sufficient information on implementation details, qualitative examples and analysis of programs and prompts.


**Weaknesses:**

* Missing error analysis or categorization on the types of examples that VPGen fails to produce accurate images (both when the layout generator fails, as well as when GLIGEN fails would be useful).

**Questions:**

* Does the domain of the image-text data used for fine-tuning the model in VPGen matter at all (for example if it were synthetic images vs real world images)?

**Limitations:**

Yes

---

> ### Author Rebuttal · Authors · 2023-08-09
>
> Thank you for the useful feedback and for pointing out our strengths of providing a strong interpretable T2I model, providing a much-needed interpretable T2I evaluation method, and having a well-written and detailed paper. We hope to address your questions and concerns below.
>
> **W1. Additional error category analysis.** Thanks for the suggestion, this is a good idea. We additionally investigated the failure cases of layout generation and show two error categories: 1) generating objects unspecified from the text prompt, and 2) not generating some objects mentioned in the text prompt. We show the error analysis results in Figure D in the attached PDF. We will include these examples/categorizations in the paper.
>
>
> **Q1. Training Domain Impact.** Our layout generation module (Vicuna 13B) in VPGen is trained on a mix of both simulated (PaintSkills dataset) and real-world images (MSCOCO/Flickr30K). We conduct an ablation study to determine if there is a difference between simulated/real-world images. We train the Vicuna on only PaintSkills and only Flickr30K and compare their results on both skill-based and open-ended prompts. For skill-based prompts, PaintSkills training shows a slightly higher average accuracy (46.8) than Flickr30K training (44.8). For open-ended prompts, PaintSkills training gets a VPEval score of (66.4), lower than Flickr30K training (71.1). This indicates that training on synthetic data can inject knowledge of specific skills such as counting and spatial relation understanding, but it is hard to cover diverse objects and attributes only with synthetic data. We will add these interesting findings to the paper.

---

### Official Review · Reviewer_1fZw · 2023-07-26

**Soundness:** 3 good
**Presentation:** 3 good
**Contribution:** 3 good
**Rating:** 5
**Confidence:** 4

**Summary:**

This paper proposes a text-to-image (T2I) generation approach along with a new evaluation framework. This can be summarized in Figure 1. First, VPGen (Sec. 3) breaks T2I down into 3 steps, with an LM to generate a “program” of objects and layouts that is then fed into the final generation module. Second, VPEval (Sec. 4) is an interpretable evaluation for T2I that invokes diverse visual modules to evaluate the generated image and the prompt’s components.

**Strengths:**

S1: Both VPGen and VPEval are sound. It leverages existing tools to make the T2I process more controllable (via layout) and interpretable.

S2: The paper is well-written.

S3: Solid experiments. The paper compares various image generation models (Table 1) and provides results based on particular skills (skill-based prompts) and their combination (open-ended prompts). Further, the paper shows a good correlation of VPEval to human judgments. Though for this point, I do have a concern since the VPEval is “gamed” to showcase VPGen (see Weaknesses).



**Weaknesses:**

W1: It is unclear how generalizable/robust the proposed T2I approach and evaluation protocol are to “non-canonical” prompts shown in Figure 1. For VPGen, what if the prompts are really complex with multiple objects that interact with one another? What if the prompts do not include the numbers of objects (e.g., Pikachus on the table)? Similarly, since VPEval is geared toward showcasing VPGen, some aspects of evaluation are missing such as object attributes (L206) (e.g., a purple Pikachu and a pink Pikachu).

W2: The novelty of both VPGen and VPEval should be highlighted further. Perhaps expand L99-105 in more detail and demonstrate L101-102 empirically. Furthermore, it would also be appreciated how the VPEval is compared to VQA-based approaches (L35) like TIFA and SeeTRUE (https://arxiv.org/abs/2305.10400), which in my opinion are also interpretable.



**Questions:**

N/A

---

> ### Author Rebuttal · Authors · 2023-08-09
>
> Thank you for the useful feedback and for pointing out our strengths - a well-written paper and solid experiments. We hope to address your concerns below.
>
> **W1-1. Challenging/Complex prompts.** Please see the general response.
>
> **W1-2. Prompts without an exact number of counts.** Thanks for the suggestion, users can indeed provide prompts without specific numbers. We find that VPGen can successfully generate layouts/images even when prompts do not have explicit counts. As shown in Figure B (e) in the attached PDF, our VPGen generates two Pikachus from the prompt “Pikachus on the table”.
>
> **W1-3. Does VPEval check attributes?** Yes. While VPEval’s skill-based prompts do not have a specific split for ‘attribute’, our open-ended evaluation prompts do include prompts that have object attributes (e.g., “A red motorcycle parked by paint chipped doors.” has attributes ‘red’, ‘parked’, and ‘paint chipped’). In open-ended evaluation, the object attributes are usually evaluated with the VQA module. Figure 4 in the supplementary pdf also shows examples with attributes. For the mentioned example prompt “a purple Pikachu and pink Pikachu”, our program generator LM (ChatGPT) generates a program that checks for the pink and purple color attributes, as shown in Code A below.
>
> Code A. VPEval generated evaluation modules for the prompt “a purple Pikachu and pink Pikachu”.
>
> ```python
> objectEval(image, 'Pikachu')
> vqa(image, 'is there a purple Pikachu?', 'yes,no', 'yes')
> vqa(image, 'is there a pink Pikachu?', 'yes,no', 'yes')
> countEval(objDet(image, 'Pikachu'), '==2')
> ```
>
> **W2-1. Baseline (closed-vocab layout generator).** We will expand the discussion of text-to-layout-to-image generation (L99-105) with some more details. Regarding L101-102 _“However, the previous approaches train a new layout predictor module from scratch, so they are limited to predicting layouts with a predefined number of classes and cannot place new objects unseen during training"_, we experiment with a layout generator with predefined 80 object classes of COCO (unlike our open-vocabulary LM Vicuna 13B). As shown in Figure A (d) in the attached PDF, the closed-vocabulary layout generator fails to generate layouts with the unseen object “Pikachu” and guides GLIGEN with the wrong layout (Note: The Pikachu in the background is generated by GLIGEN, and there is no layout bounding box for the Pikachu).
>
>
>
> **W2-2. VPEval compared to TIFA / SeeTRUE.** While both TIFA/SeeTRUE are also based on atomic questions and thus considered interpretable, the main difference between TIFA/SeeTRUE and our VPEval is the adoption of diverse evaluation models. While TIFA/SeeTRUE rely on a single evaluation module (VQA) that cannot cover different T2I skills, our VPEval adopts diverse evaluation models, including object detection, counting, OCR, etc. Sec. 5.4 shows that our diverse evaluation modules of VPEval achieve higher human correlation than TIFA on both skill-based and open-ended prompts. VPEval can also provide visual explanations (in the form of bounding boxes) and error messages, whereas TIFA/SeeTRUE cannot. Please also note that the SeeTRUE paper appeared on Arxiv on May 17, 2023, the same date as NeurIPS 2023 full paper submission deadline; we will cite SeeTRUE in the paper.

---

### Official Review · Reviewer_aEEd · 2023-07-27

**Soundness:** 3 good
**Presentation:** 3 good
**Contribution:** 3 good
**Rating:** 6
**Confidence:** 4

**Summary:**

This paper delves into the intersection of large language models (LLMs) and their applications in vision-and-language tasks, specifically focusing on text-to-image (T2I) generation. The authors identify a gap in the existing literature: no thorough analysis has been conducted on the synergy of LLMs and various visual modules for "complex" T2I generation tasks.

To address this they propose two novel frameworks: VPGEN: This framework offers a step-by-step approach to T2I generation, segmenting the process into three distinct stages: object/count generation, layout generation, and image generation. They use Vicuna to fine-tune to manage the first two stages, and the results demonstrate improved control over layout creation. The final stage, image generation, incorporates existing models like GLIGEN. Notably, VPGEN’s design capitalizes on the inherent knowledge of pre-trained LLMs, granting it the ability to recognize objects that have not been predefined—surpassing the capabilities of older layout-guided T2I techniques.
VPEVAL: This authors propose this for evaluation of the generated images. Unlike conventional T2I evaluation methods which primarily gauge visual quality and image-text alignment through a singular visual module, VPEVAL emphasizes interpretability and a multiple modules to evaluate the generated image. In essence, it employs evaluation programs that activate a variety of visual modules, to distinct T2I skills.
The papers findings indicate that VPGEN+GLIGEN combination showcases relatively better performance, especially when precision in layouts and spatial relationships is paramount. The VPEVAL evaluation method seems to aligns with human assessment.
In summary the contributions are:
The introduction of VPGEN, an interpretable T2I generation framework that dissects the T2I process into three modules.
The proposal of VPEVAL, an evaluation framework for T2I tasks that enhances the explainability and thorough analysis by invoking diverse visual modules.
A detailed analysis of various T2I models, highlighting the superior layout control of VPGEN and the human-centric alignment of VPEVAL.


**Strengths:**

Overall its a great work and the paper addresses some fundamental limitations in a unique way.
The paper addresses a very important problem in image generation realm. Most of the existing image generation models including the ones trained on massive datasets, are not good at generating images with spatial consistency, generating the correct number of objects , or understanding the size of the objects. This bias has been highlighted by various papers since 2018 and is still existing in modern generative models. The paper proposes a solution to the same by combining LLMs fine tuned on text to layout pairs and layout to image generation model. This fine tuning provides the LLMs with the ability to understand the spatial etc relationships between the generated objects. Thus removing a limitation from the prior results.
The proposed VPEval combined several modules to evaluate the generated quality of the images and also interpretable due to the generated programs. Using such visual programming is novel.
Adaptability - as the modules in image generation and image evaluation improve the proposed approach would also improve in performance.
Offering visual+textual explanations is a strength as it increases the interpretability of the evaluation.

**Weaknesses:**

The proposed 5 evaluation modules might still not capture the generated image quality and other complex semantics. Other image quality could be incorporated and evaluated.

The paper emphasizes heavily on the two step generation - object count and layout generation , but fails to conduct any ablation studies to prove the effectiveness of this pipeline. Since the process is decomposed into different steps, the interplay and coherence between these steps are vital. Errors or inconsistencies in earlier steps (like object/count generation) could cascade and affect the final image's quality.

Relying on bounding boxes to represent layout may be simplistic. The boxes encode basic spatial relations for sure but might miss more detailed pose, occlusion, and depth information.

Associating object names in texts to layouts could be unreliable for ambiguous or synonymous words. The model may lack grounding to map words to visual concepts.
For example, consider the word "bat" - this could refer to: A baseball bat A flying bat animal
Or the word "apple" could refer to: The fruit apple The technology company Apple
Without proper grounding between language and vision, the model may struggle to determine which visual concept is being referred to based on just the word alone.
Some detailed examples to illustrate this:
The text "a man holding a bat" is ambiguous - is this referring to a baseball bat or a flying animal bat? The model may wrongly depict it without the proper grounding. The text "a logo of an apple" could wrongly depict the fruit when the technology company was meant. Synonyms like "couch" vs "sofa" would need to be mapped to the same visual concept. So in summary, relying purely on language without grounding it properly to visual concepts can lead to ambiguity in mapping words to the intended visual representations. Providing more context and grounding is important to resolve this.

Granularity of Layout Representation: VPGEN decomposes bounding box coordinates into a [0,1] scale and quantizes them into 100 bins. Such discretization can potentially lead to a loss of finer details in bounding box representation, which might impact the accuracy of object placement in the generated images.

2 human evaluators are slightly less to conclude that the results align with human and also the setting of human evaluation is a bit unclear. Like humans evaluate it subjectively (esp. with open ended tasks) . So could the authors shed some light on this setting? Also given that the correlation doesnt seem very high for objects ( 63.7?)

Minor comments:
The word skill has been mentioned earlier without much context which might be ambiguous to the readers ( so just a minot presentation suggestion) .
Explaining what the paper means by "challenging" T2I tasks might be helpful in terms of better clarity and presentation.
Scaling the approach to generate complex high-resolution images with many objects and intricate relations may be difficult. The text-to-layout-to-image pipeline has limitations. Any further results to prove or disprove this might be helpful.


**Questions:**

1. How would the authors compare their approach with Visual chatgpt ( https://arxiv.org/pdf/2303.04671.pdf) . This seem to have a image generation module?

2. Does quantizing the bounding box coordinates into discrete bins loses precision? does this lead to misalignment between the predicted layout and actual image content.

3. Unseen Objects: The ability to generate layouts of objects not seen during training (e.g., 'pikachu') can be an advantage, but it also raises questions about the model's ability to accurately represent unfamiliar objects in space. How well can it handle a completely novel entity in terms of spatial characteristics, especially when combined with other objects?

4. How many in-context examples are used in the for the VPEval task?

5. Can the authors provide results for count greater than 4 ( fig 5) . and can the model generate images like " generate a circle with 3 cm radius and generate a circle with 6 cm radius , essentially saying one smaller than the other ?

6. The model has a limited encoding capacity to represent all possible object combinations and numbers. So  to understand if an unusual or out-of-distribution combinations at test time could confuse the layout prediction stage, can the authors provide some complex compositional image generation results?

7. what is the inference time?

**Limitations:**

The authors have spoken about the limitations.

---

> ### Author Rebuttal · Authors · 2023-08-09
>
> Thank you for the useful feedback and for pointing out our strengths in addressing important issues in the image generation community regarding spatial control and having an adaptable/interpretable evaluation framework.
>
> **W1. Image quality metric (FID).** Please see the general response.
>
> **W2. Error propagation analysis.** We analyze the performance of each step in the VPGen pipeline to determine how much error gets propagated.
>
> 1. _Object Recall_: The ratio of whether correct objects are included in the generated layouts.
> 2. _Object Count_: The ratio of the correct number of objects is included in the generated layouts.
> 3. _Layout Accuracy_: VPEval accuracy of the generated layouts (using the generated layouts as the object detection results).
> 4. _Image Accuracy_: VPEval accuracy of the final image by GLIGEN (in Table 1).
>
> As shown in Table B, the layout accuracy is much higher than image accuracy, especially for Count/Spatial skills. This indicates that the major error source is the image rendering step (with GLIGEN), and using a more accurate layout-to-image generation model in the future would improve the accuracy of the VPGen pipeline. We will incorporate the results in the paper.
>
> Table B. Step-wise error analysis of VPGen. For the ‘Spatial’ skill, “front” and “behind” splits are skipped in this table since Vicuna13B does not generate depth information.
>
> | Skills  | Vicuna13B Object Recall (%)  | Vicuna13B Object Count (%) | Vicuna13B Layout Accuracy (%) | Layout-to-Image (GLIGEN) Accuracy (%) |
> |---|:---:|:---:|:---:|:---:|
> | Object | 99 | 99 | 99 |  97 |
> | Count | 99 | 99 | 99  |  72 |
> | Spatial | 98 | 98 | 88 | 34  |
> | Scale | 93 | 93 | 38 |  23 |
>
> **W3. Bounding box layout.**  While we can also guide image generation models with pose keypoints/segmentation/depth map, in VPGen, we choose the bounding box layout format because of its efficiency; bounding boxes require a much smaller number of tokens to represent compared to other formats (e.g., a xyxy-format bounding box can be represented with 4 tokens; while a 64x64 segmentation map requires 4096 tokens).
>
> Please also note that our VPGen framework can be extended to pose/occlusion/depth guidance when we find an efficient way to generate them with LM and have access to an image generation model that can take such information as input.
>
> **W4. Words can be ambiguous.** We agree that the text prompts do not always have detailed information. While we focus on introducing the first VP framework for text-to-image generation, we suggest several ways to address the ambiguity below. As these ideas are involved with new method design/data collection/experiments, we leave this to future work.
>
> 1. Multi-turn interaction
> ```
> User: “Draw a bat”
> System: “The word ‘bat’ is ambiguous, do you mean an animal or a baseball bat?”
> User: “animal”
> ```
> 2. Show an exemplar image using a multimodal LM
> ```
> User: “Draw a bat like <image>”
> ```
>
> **W5 & Q2. Granularity of layout.** We follow previous LM-based object detection to normalize/discretize the boxes [48, 49] (L137). In Table C, the 100 and 1000 bin settings show almost identical accuracies, suggesting that the current 100 bin discretization does not hurt the accuracy of object placements. We will add the results to the paper.
>
> Table C. 1000 vs. 100 bins.
>
> | # Bins  | Avg. VPEval skill-based Acc. | VPEval open-ended Acc. |
> |---|:---:|:---:|
> | 1000 | 47.8 | 69.1 |
> | 100 | 47.0 | 70.0  |
>
> **W6. Human eval setup.** For open-ended eval, we borrow the human evaluation scores (with two annotators) from TIFA [35]. Following TIFA, we use two human annotators for skill-based prompt evaluation as the prompts are short and straightforward (e.g., “two dogs”). Our annotators achieve high inter-annotator agreement with Cohen’s $\kappa$ and Krippendorff’s $\alpha = 0.85$ (> 0.8 indicates near-perfect [61; 62; 63], L304-306). Also, note that Spearman’s $\rho=63.7$ of VPEval-Object is higher than other methods in Table 3.
>
> **Minor comments - presentation suggestions.** Thanks for the suggestions. We will explain/clarify ‘skill’ and ‘challenging’ in the paper.
>
> **Q1. Comparison to Visual ChatGPT.**  Visual ChatGPT directly calls a T2I model API (SD) with a given prompt to generate an image. In contrast, in our VPGen, LM has a more important role of semantic layout parsing with two steps: 1) generating object counts and 2) generating layouts given object counts.
>
> **Q2. Precision loss from discrete bins?** We find that using bins does not impact performance. Please see the answer to “W5 & Q2. Granularity of Layout Representation.”
>
> **Q3. Unseen object spatial control.** As suggested, we generate images with a series of prompts to see how well it handles placing unseen objects (Pikachus / Link) in different spatial relations (on a table / holding). As shown in Figure A (a,b,c) in the attached PDF, our layout generation LM can place the unseen objects in the correct locations.
>
>
>
> **Q4.  In-context examples used for VPEval.** For a fair comparison with TIFA [35], we follow the same 12 prompts used in TIFA’s question generation for in-context examples for VPEval program generation. We will include this detail in the paper.
>
> **Q5-1. Counts > 4.** We show the count skill results with 5, 6, and 7 in Table D. As expected, the trend continues from Fig 5, where higher counts are more challenging. We show the generation examples in Figure B (a,b,c,d) in the attached PDF. We will add the results to the paper.
>
> Table D. ‘Count’ accuracy of VPGen.
>
> | Counts |  1 |  2 |  3 |  4 |  5 |  6 |  7 |
> |---|:--:|:--:|:--:|:--:|:--:|:--:|:--:|
> | VPEval Count Accuracy (%) $\uparrow$ | 85 | 80 | 66 | 55 | 48 | 35 | 28 |
>
>
>
> **Q5-2. & Q6. Challenging/complex prompts.** Please see the general response.
>
>
> **Q7. Inference time.** On a single A6000 GPU, VPGen takes around 6s (2s for Vicuna 13B + 4s for GLIGEN) to generate an image and VPEval takes <1s (typically <0.5s) per evaluation module. We will add this information to the paper.

---

> > ### Comment · Reviewer_aEEd · 2023-08-21
> > **Acknowledging Authors Rebuttal**
> >
> > I've carefully reviewed the authors' responses and I recognize the clarifications provided on my concerns. Their insight on "W4: Word Ambiguity" is particularly interesting. However, considering the potential for increased prompt complexity, such as indicated in the attached PDF referencing "generating unseen objects a, b", having only two human evaluators might fall short of ensuring comprehensive assessment. The reported inference times are commendable. I concur with the authors' perspective that expanding the dataset or fine-tuning methodologies might bolster the spatial accuracy of generated images ( big circle vs small circle radius). While there might be other potential other avenues to enhance this, dataset scaling indeed appears to be a viable solution. On the whole, this paper is great in its approach to addressing a fundamental challenge in a creative way. I enjoyed reading the paper, and the authors' dedication to the subject matter is great.
> >
> > I am keen to read any updated version to observe the incorporated changes.

---

> > > ### Author Response · Authors · 2023-08-21
> > >
> > > Thanks for your supportive and positive discussions. We are glad that you enjoyed reading the paper and appreciated our clarifications. We will add these additional details in the final version.

---

### Author Rebuttal · Authors · 2023-08-09

We thank the reviewers for their valuable feedback and for recognizing our strengths:
- addressing an important/foundational problem in text-to-image generation (aEEd)
- developing strong/interpretable/useful text-to-image generation and evaluation frameworks (aEEd, 1fZw, GZ2e, PtVL, bJid)
- providing better spatial controllability for text-to-image generation (aEEd, 1fZw, PtVL, bJid)
- having generalizable/scalable frameworks (aEEd, GZ2e)
- providing solid experiments (1fZw, GZ2e, PtVL)
- and being well-written (1fZw, Gz2e).

**Attached figure-PDF**

In the attached PDF, we include visualizations that cover
- unseen objects (Figure A)
- counting beyond 4 (Figure B)
- challenging/complex prompts (Figure C)
- and layout error categorization (Figure D)

to address your comments.

**Common Answers**

**Reviewer aEEd W1. and bJid W1. - image quality metric (FID).** Thanks for the suggestion. In Table A, we compare our VPGen (Vicuna13B+GLIGEN) to its backbone Stable Diffusion (SD) v1.4, in FID (30K images of COCO val 2014 split) as well as VPEval Acc. Both VPGen checkpoints show better Skill-based Acc than SD v1.4 while achieving comparable Open-ended Acc. and FID.
In FID (lower the better), we find VPGen (Flickr30k) < SD v1.4 < VPGen (Flickr30k+COCO+Paintskills). We think a bit of increase (but still reasonably good) in the FID of the “Flickr30k+COCO+Paintskills'' checkpoint is because the layouts of PaintSkills are different from those of natural scenes (COCO and Flickr30k). We will add the results to the paper.

Table A. VPEval accuracy and FID.

| Model  | VPEval Skill-based Acc. (%) $\uparrow$ | VPEval Open-ended Acc. (%) $\uparrow$ | FID (COCO 30K) $\downarrow$  |
|---|:---:|:---:|:---:|
| SD v1.4 | 37.4   | 70.3  | 16.5 |
| VPGen (Flickr30k+COCO+Paintskills) | **47.0**   | 70.0  | 20.1 |
| VPGen (Flickr30k) | 44.6   | **71.2**  | **15.9** |


**Reviewer aEEd Q5-2. & Q6. and Reviewer 1fZw W1-1. Challenging/complex prompts.** As requested, we show images generated with the complex prompts (1) “A realistic photo of a Pomeranian dressed up like a 1980s professional wrestler with neon green and neon orange face paint and bright green wrestling tights with bright orange boots” (from DrawBench [24]) and (2) “a circle with 3 cm radius and a circle with 6 cm radius”, in Figure C (a,b) in the attached PDF.

VPGen understands the important parts of the prompts (e.g., generating ‘realistic’, ‘Pomeranian dog’, bright orange boots’; generating layouts of two circles in different sizes), but misses some aspects (e.g., the bigger circle is not twice the size of the smaller circle). This is probably because the training prompts (COCO, PaintSkills, Flickr30k) do not include many prompts written in these styles. We believe that scaling datasets with diverse sources can further improve VPGen.

---

### Decision · Program_Chairs · 2023-09-21

**Decision:**

Accept (poster)

**Comment:**

The work received positive reviews. The reviewers found the approach simple and novel and the paper to be well written with good experiments. The rebuttal successfully addressed most of reviewer’s concerns. Among the negatives, the reviewers wished to see further error analysis and expressed some concern about the robustness of VPEval.

The AC agrees with the overall positive sentiment and believes the community would benefit from further exploring ideas such as VPEval which involve programatic verification of T2I models. VPGen results also indicate a greater degree of control over layout and semantics compared to existing literature. Hence the AC is happy to recommend acceptance.

The AC requests the authors to incorporate the suggestions from reviewers and any new results presented during author-reviewer discussion. Congratulations!